# A Framework for Operational Management of Urban Water Systems to Improve Resilience

Jorge Cardoso-Gonçalves * and José Tentúgal-Valente

Department of Civil Engineering, Faculty of Engineering, University of Porto, 4200-465 Porto, Portugal; tentugalvalente@gmail.com
* Correspondence: jjtc.goncalves@gmail.com

**Abstract:** Optimizing the management of hydraulic infrastructures that support water supply, wastewater, and stormwater drainage can increase the efficiency of these systems. A framework for operational management of urban water systems allows for robust management, which contributes to the system's overall resilience. A methodology has been structured to support the decision-making process of managing entities. The methodology for the operational management of hydraulic infrastructures incorporates concepts of asset management, risk management, and technical management. It is organized into three operational areas (assessment, operation, and intervention) and aims to increase the efficiency of managing entities. Two cases were used to implement the aforementioned methodology—the Arouca Water Supply System (SAA-Arouca) and the Trofa Wastewater Drainage System (SAR-Trofa), both under the responsibility of Águas do Norte, S.A. In SAA-Arouca. There was a particularly significant reduction in the system input volume (purchased the first level) and the number of pipe busts observed in the subsequent period after the implementation of the methodology. Regarding the SAR-Trofa, the application of the methodology focused particularly on improper inflows (rainwater and others). The proposals for this system mainly aim at reducing the volumes collected by the drainage networks (in low-level infrastructures) and delivering them to different high-level infrastructures.

**Keywords:** operational management; hydraulic infrastructures; water systems' resilience; water supply; water losses; wastewater drainage; inflows

## 1. Introduction

The demographic evolution in the second half of the 20th century, particularly in the 60s, leveraged the construction of housing estates and public facilities, rising to the expansion of hydraulic infrastructures [1]. Currently, the increase in urbanization in concentrated hubs and the desertification of rural areas, the improvement of service and increased demand, the conclusion of new systems in a significant number of locations, the operation of aging systems, the assurance of performance levels with increasing efficiency, scenarios of resource scarcity, the strong availability of data and technological means to support management, and the need to preserve existing know-how and empower new human resources are aspects of the reality observed in the current context of hydraulic infrastructure management. On the one hand, this demonstrates the overall positive balance of recent decades. On the other hand, it highlights the need to reflect on current vulnerabilities and opportunities [2].

The increase in population and the consequent increase in consumption, in the context of climate adaptation, have been accentuating water scarcity in different regions of the world [3–6]. Currently, urban areas are consolidated and fully or partially infrastructured. The challenge for the management of hydraulic infrastructures is identified in this context: the renovation and adaptation of systems [7].

Alegre et al. [8] (2012) state that public water services represent a paradigmatic example, emphasizing that they are essential public services, although commonly undervalued. The supporting infrastructures are of limited visibility and entail high construction costs and durability. Using water efficiently is to promote processes and technologies without compromising the efficacy. Some measures that can be considered, regarding public systems of water supply and water drainage are: the optimization of opportunities and processes, pressure and water losses reduction in supply systems, adequate taxes, and reusing treated urban wastewater [9].

The GPI (asset management) structures traditional management practices, integrating them into principles of objective management and continuous improvement and favoring new techniques of analysis, comparison, and communication. GPI balances performance, cost, and risk based on management, engineering, and information, planned strategically, technically, and operationally. It is mentioned that defining priorities and selecting intervention needs require knowledge of assets (infrastructures), and diagnosing the existing situation allows the assessment of residual life and the economic value of infrastructure [10]. An organic approach of asset management must include an analysis to rehabilitation and development of plans. These can be reinforced by evaluating interventions and considering alternatives [11].

Planning the rehabilitation of aged infrastructures needs a critical analysis on two main questions — How much investment is needed to maintain a sustainable service? – and — Which of the components are the priority? It should be noted that although the issues are closely linked, the analysis approaches are different, with the first relating to strategic infrastructure rehabilitation planning, which seeks to establish long-term investment strategies, and the second to medium-term tactical planning [12–14].

In the decision-making process, risk is approached as the inability to anticipate the structure, results, and consequences, including uncertainty and its outcomes [15]. Including risk in the decision-making process requires considering two factors: the probability of scenario occurrence and the probable consequences of scenarios. A decision that accounts for these is referred to as a decision informed by risk [16]. Regarding the technical applications of quantitative risk analysis, the general technical definition of risk is considered sound, starting from the hypothetical occurrence of an event with negative or positive consequences. Risk is determined by the product of probabilities and consequences [16]. It is suggested to use the general technical definition of risk because it is an expedient calculation methodology that can be based on real data and allows the use of risk from an operational, adaptive, and versatile perspective, informing decision-making. The efficient management of water distribution systems and the potential associated with telemetry systems as generators of consumption data are challenges related to a more in-depth knowledge of urban consumption. The use of this data in engineering applications is a long way off, given the demands that these applications place on the data collected [17].

Structurally, Almeida [18] (2017) mentions that most managing entities do not have organizational documents (such as water safety plans, loss reduction plans, or meter management plans) and emphasizes that experts identify these documents as one of the main details of "organizational causes", referred to as the second main cause of water losses, just after "infrastructural causes".

The proposed methodology was developed by Jorge Cardoso Gonçalves. The Operational Management of Hydraulic Infrastructures methodology (GOIH, in Portuguese) is presented as an aggregating element of asset management, risk management, and technical management concepts, proposing a new organizational model detailed further in Section 2.2 in order to tackle the referred vulnerabilities and take advantage of the opportunities.

Water must be managed resiliently when responding to the challenges emerging from the exploration of water supply, wastewater, and rainwater drainage systems—water consumption and water treatment before discharge into receiver water bodies are matters to consider [19]. Urban and industrial development is demanding natural resources. Hence, it is necessary to promote water management policies to ensure the renewal of

water resources, which can benefit future generations [20]. The efficiency of the water management sector depends on daily proactive measures [21]. A proper management policy, along with the application of the most effective techniques, is the key to the success of this process. The positive results associated with water losses in supply systems are proportional to the economic and financial resources [22].

GOIH, presented in this article, aims for optimized management of water systems (supply and/or drainage), focusing on economically viable and environmentally sustainable solutions. Preservation of water sources, availability in scarcity scenarios, supply security, protecting the receptor water bodies, response to extreme phenomena, and adaptation to new contexts are some of the issues considered in this methodology [19]. This methodology is based on three main steps in order to reach the final proposed solutions to the managing entities: data analysis, the methodology itself (evaluation, exploration, and intervention, among other matters), and decision support, which are divided into several processes, as shown in Figure 1 [23].

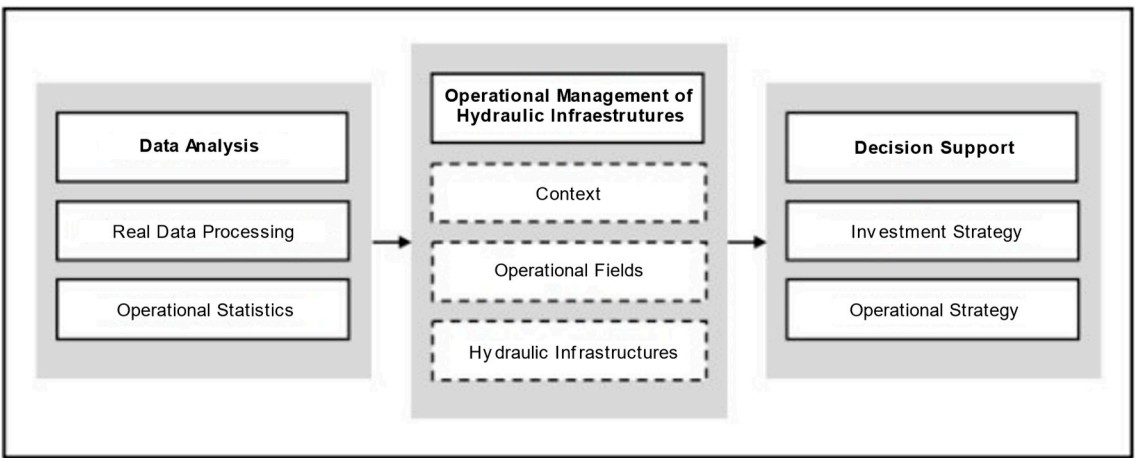

**Figure 1.** Organizational structure of the framework (adapted from [23]).

## 2. Operational Management of Hydraulic Infrastructures (GOIH)

Operational Management of Hydraulic Infrastructures is based on the management concepts of patrimony management—state, performance, costs, and risks; risk management—simplified calculation methods, real data, and management informed by risk; and technical management—operational and organizational concepts. It also considers the inherent challenges of the water service's management and the supporting hydraulic infrastructure. The methodology aims for the development of strategies that consider the environment (searching for environmentally sustainable solutions), society (considering people's need for effective water supply and wastewater and rainwater drainage), the economy (searching for efficient solutions with fewer financial resources that stimulate the market), and politics (involving decision-makers). It also considers science and innovation when trying to bring the academic community and managing entities closer and develop new solutions.

The GOIH methodology considers the challenges of the water system's management, which include an integrated perspective of the sector and society's several dynamics, the resilient management of water resources and their sustainable usage, adaptation to climate change consequences, and operational optimization. All of the proposed solutions must respond effectively to these challenges.

This methodology depends on political support, effective leadership, clear and attainable goals, and enough time to implement the proposed solutions. It also needs cross-border and broad communication with all the involved parties.

After establishing the context of the systems in study, further steps of the methodology take place (Figure 2).

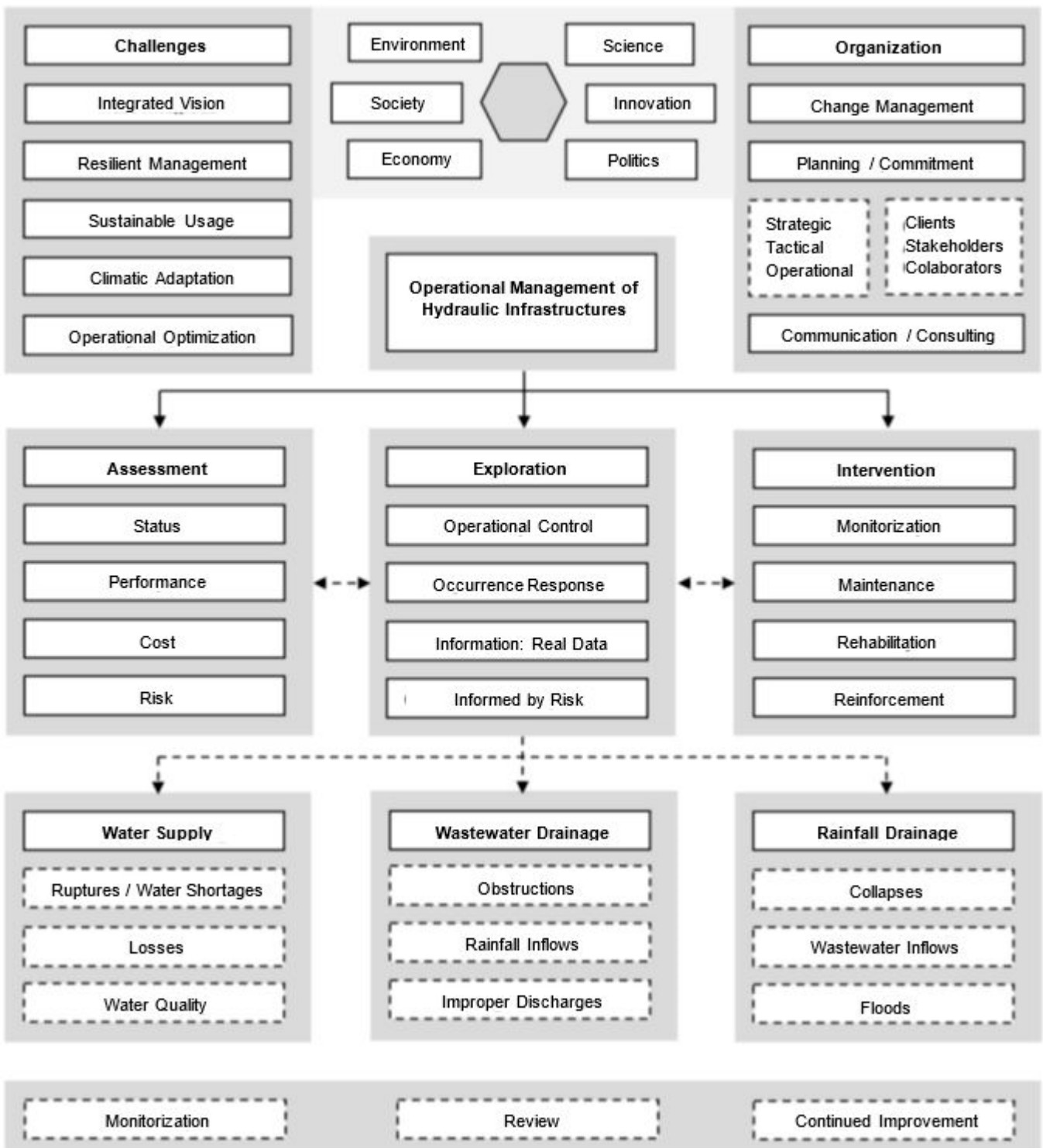

**Figure 2.** Organizational structure of the Operational Management of Hydraulic Infrastructures (adapted from [23]).

According to Figure 2, after context is established, the methodology follows the operational areas—assessment, exploration, and intervention.

### 2.1. Data Analysis

The process of data analysis (Figure 1) is sorted between the processing of real data and operational statistics.

Regarding the processing of real data, it starts with a primary analysis to identify the main peculiarities and eventual abnormal data—descriptive analysis. Following this comes *data systematization*, where data are sorted between data from the managing entity and data obtained throughout the present study. The data are also organized into three main categories:

**Infrastructure data**—concerning infrastructure characterization and inquiries into data regarding the state, performance, and risk of the infrastructure.

**Exploration data**—data that pertains to monitoring, occurrence records, intervention logs, and other operational data.

**Management data**—management reports, the system's performance, and commercial information.

The detection and deletion of abnormal data are carried out throughout the whole process, after the descriptive analysis and data systematization. Data normalization allows us to unify the collected information, which in turn allows for a comparison between the different data sets. Data reduction makes a long period of analysis possible by compacting the initial data volume.

After the real data has been processed, the statistical treatment of technical information is carried out—operational statistics. An empirical and rational approach is chosen over more sophisticated mathematical models or probabilistic models for scenario simulations. This step—operational statistics—consists of descriptive statistics, information organization, and a risk approach.

Descriptive statistics consists of the analysis of some statistical elements obtained from the collected data, such as the mean, standard deviation, range, and mode, among others.

When organizing the information, the collected and processed data should be converted into tables and graphs.

The risk approach is flexible, with variable complexity sustained in the data provided by the managing entity, to estimate the risk and support the decision-making process. The risk degree of an event is directly correlated with the probability of its occurrence and its consequences. The probability and consequence calculation as well as risk estimation can be sustained in three types of approaches:

**Objective approach**—sustained by real exploration data. A quantitative analysis of the probability of event occurrence is carried out through a ratio between the number of occurrences of one system and the total number of occurrences. As for the consequences of occurrences, these are measured according to the monetary consequences of the events. Lastly, risk is determined by the product of probability of the occurrence and consequences for isolated or associated events.

**Subjective approach**—based on inquiries to the managing entity. The probability of occurrence is assessed qualitatively, while the consequences are assessed empirically. Risk is assessed using a risk matrix.

**Modelling-based approach**—sustained in hydraulics modelling and risk and reliability analysis.

Finally, the value of the calculated risk is translated into a monetary value.

### 2.2. Infrastructure Assessment

The assessment of the infrastructure is sorted into four categories: status, performance, cost, and risk. Status, performance, and risk assessment establishes five categories for each assessment—from class 1 (very good, excellent, and very low, respectively) to class 5 (very bad, inefficient, and very high, respectively). Cost assessment is based on several factors, as shown in Figure 3.

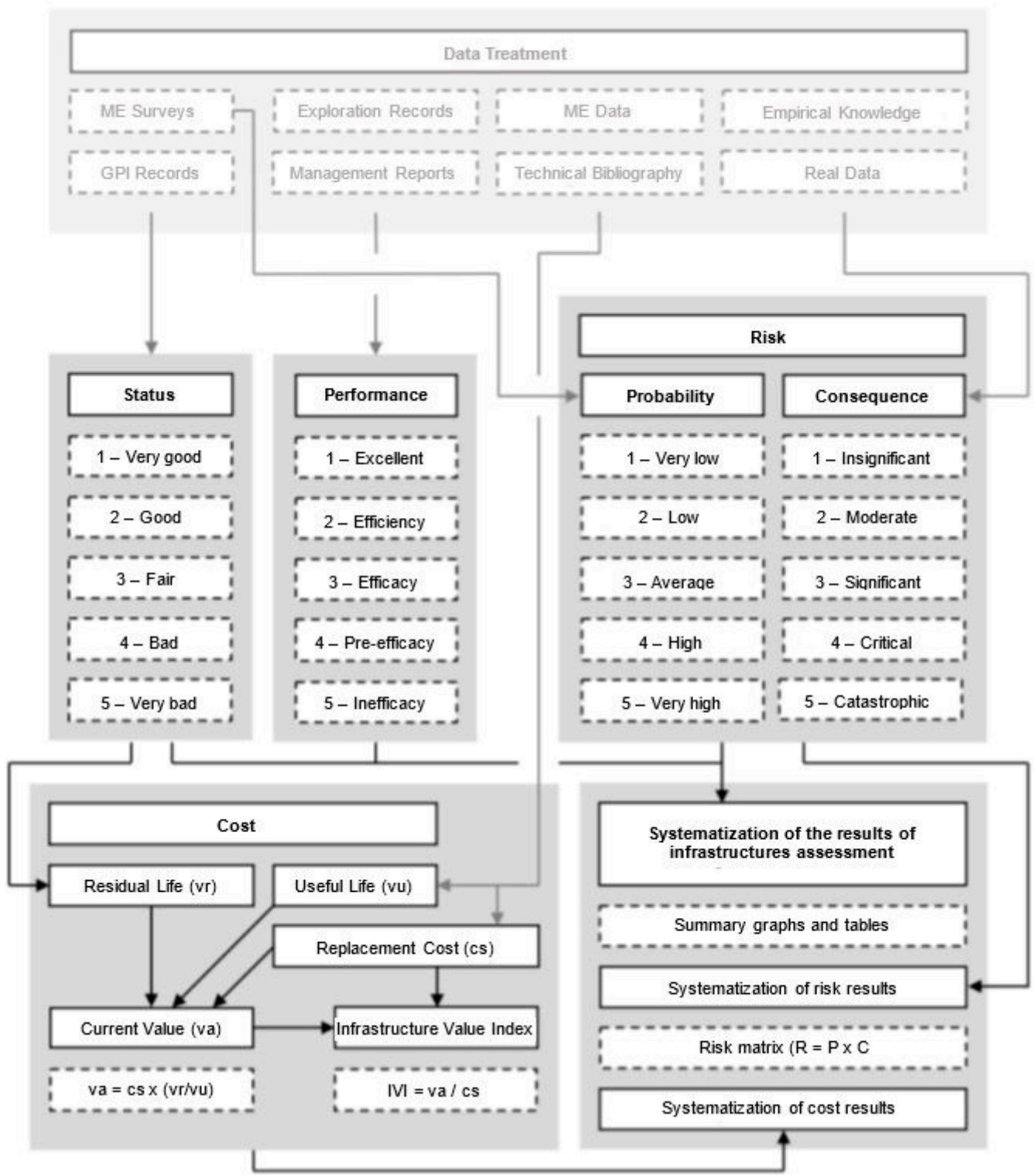

**Figure 3.** Infrastructure assessment method (adapted from [23]).

### 2.3. Exploration

This stage of the GOIH methodology is based on the data collected through the managing entities and is sorted in four steps that can be excluded or added according to the infrastructure under study:

- Operational control

This stage of *Exploration* consists of systematic recording (occurrences, monitoring data, anomalies, updates, and interventions); operating costs; human resources management (including structured operational routines, implementation of specialized teams, and action planning); and finally, real-time management (through monitoring and alert systems). Some of these are in common with Section 2.1. (Data Analysis) and Section 2.2. (Infrastructure Assessment).

- Response to occurrences

When occurrences take place, resolution depends on human resources and material means (prevention teams, external contractors, and operational storage), operational relevance index (IRO, in Portuguese), and prediction modeling for preparing preventive action.

Regarding IRO, this index is used for pre-positioning human and material means and for prioritizing operating actions. IRO translates the relevance of certain activities, infrastructure, or subsystems through the average weights of different indicators of the system.

- Information management

This focuses on the conversion of the collected information into knowledge about occurrences (number, type, time and space distribution, and locations with the most incidence) and performance information.

- Risk-informed management

This risk-informed management is based on risk calculation based on real data—calculation of probability (ration between occurrences associated with an infrastructure or subsystem and the total number of occurrences); consequences (operating costs associated with occurrences); and risk, which is the product resulting from probability and consequences. This type of management also depends on objective risk assessment and risk-informed decision-making.

### 2.4. Intervention

This stage of GOIH is divided into five categories to organize the interventions necessary in the different components of water systems, with the goal being prioritization, action planning, summarizing costs, and optimizing the operational management of hydraulic infrastructure. The four categories are as follows:

Monitorization;
Maintenance;
Rehabilitation; and
Reinforcement.

Of these four categories, Monitorization and Maintenance include the same steps: systematization of past, ongoing, and future actions; action planning (plans, routines, and actions); data summary; cost analysis; and future action definition. Rehabilitation and Reinforcement include the aforementioned systematization, investment planning, intervention summary, and future action definition.

### 2.5. Decision Support

Decision support for managing entities is associated with investment planning since financial resources must be managed carefully, directing them into solutions that result in larger gains and shorter return periods. This tool of the GOIH methodology is based on the data gathered in all the previous stages explained regarding the status, performance, cost, and risk of the hydraulic infrastructure.

Two types of investment can take place:

With operational return, it envisions performance improvement of the infrastructure. Its intervention aims to decrease occurrences and decrease outflows and inflows, among others. It is also analyzed based on its estimated return.

With strategic return, it focuses on interventions that are assessed based on the global interest of their execution from a system's evolution standpoint (to reinforce responsiveness, increase the coverage rate, or improve the service's security and quality).

The data attained in the above-mentioned stages is converted through decision support indexes—status evaluation index (IAE), performance evaluation index (IAD), risk assessment index (IAR), operational relevance index (IRO), risk management index, and context factors index (IFC). Directly resulting from the operational field of *Assessment* is cost assessment, translated into replacement costs, current value, and IVI (infrastructure value index); and resulting from the *Exploration* field are occurrence summaries and systematized operational costs.

The decision-making process will encompass a selection of interventions, the prioritization of interventions, and the estimated investment.

The decision support winds up at three levels:

- Investment strategy

This stage of strategy is based on indexes, which are based on the indexes aforementioned—the infrastructure investment index (III) and the operational investment index (IIO). This is a flexible methodology; hence, the decision does not need to use the totality of the indexes, and it may include others.

- Operational strategy

This strategy considers investments, gains, and return periods.

The annual gains are calculated considering the annual economic losses and the current economic gains. Similarly, it is also possible to calculate future economic gains. These economic losses may be deleted if a certain investment is executed.

The return period is dependent on the investment amount and the annual gain due to the investment. The higher the annual gain, the shorter the return period.

- Specific decision strategy

In this stage, the investments under analysis must not be only the monetarily measurable ones. At this stage, predictions of the evolution of the systems, medium- to long-term, can be included.

The gains associated with the chosen investments may finance a transformative process for the managing entities, along with gains for the stakeholders.

## 3. Case Studies

### 3.1. Water Supply System of Arouca

This system supplies the Arouca municipality, which has an area of 327 km$^2$ and approximately 24,000 inhabitants. Topographically, the area is rural with great altitude variations.

Regarding the supply system, the Water Supply System of Arouca (SAA-Arouca, in Portuguese) consists of a Main System (MS) and Autonomous Systems (AS). The MS is supplied through 6 Delivery Points (DP) provided by Águas do Douro e Paiva, S.A. (AdDP), and its 25 subsystems are additionally provided by 27 water catchments (information related to 2017, according to the managing entity) [23]. Figure 4 shows the distribution of the system in its Main and Autonomous Systems.

In the past, the SAA-Arouca was divided into 58 subsystems. This led to interruptions in the supply chain, especially during the summer. Because of this problem, a main pipeline at a higher level of the system was executed. With the several alterations done to the SAA-Arouca throughout, with a focus on the balance of supplied areas and the deactivation of autonomous sources, Figure 5 shows the supply areas in 2017.

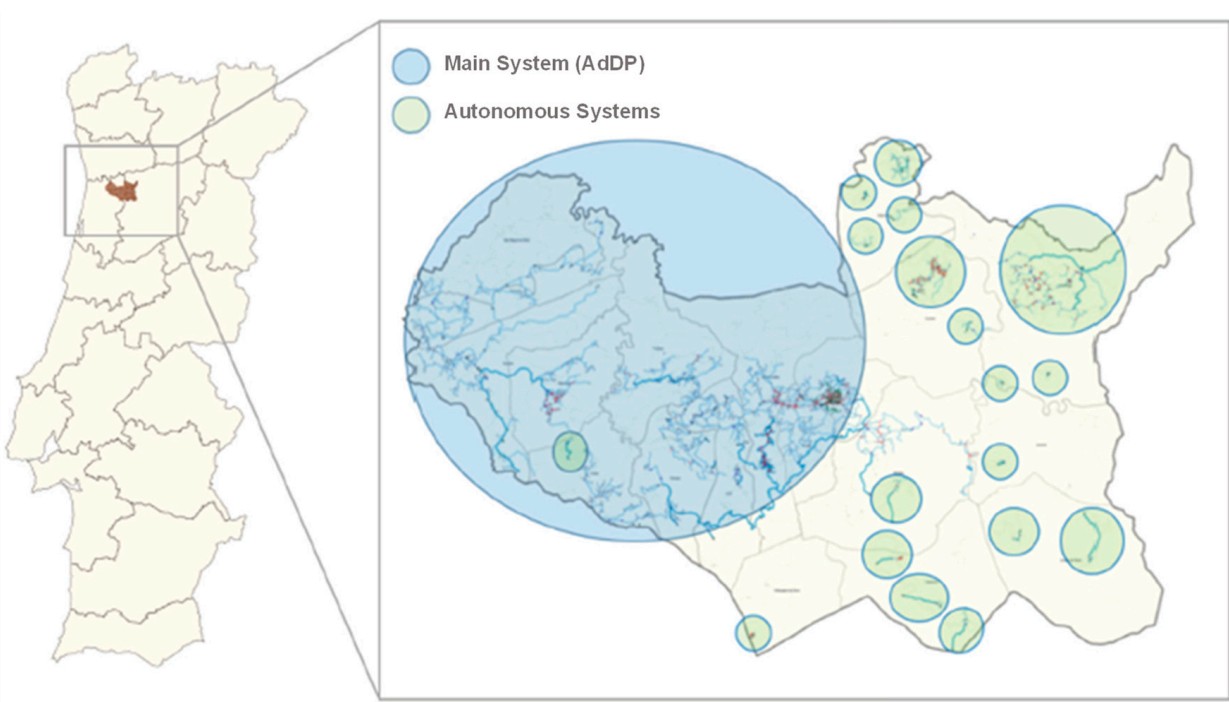

**Figure 4.** Representation of the subsystems of the SAA-Arouca (adapted from [2]).

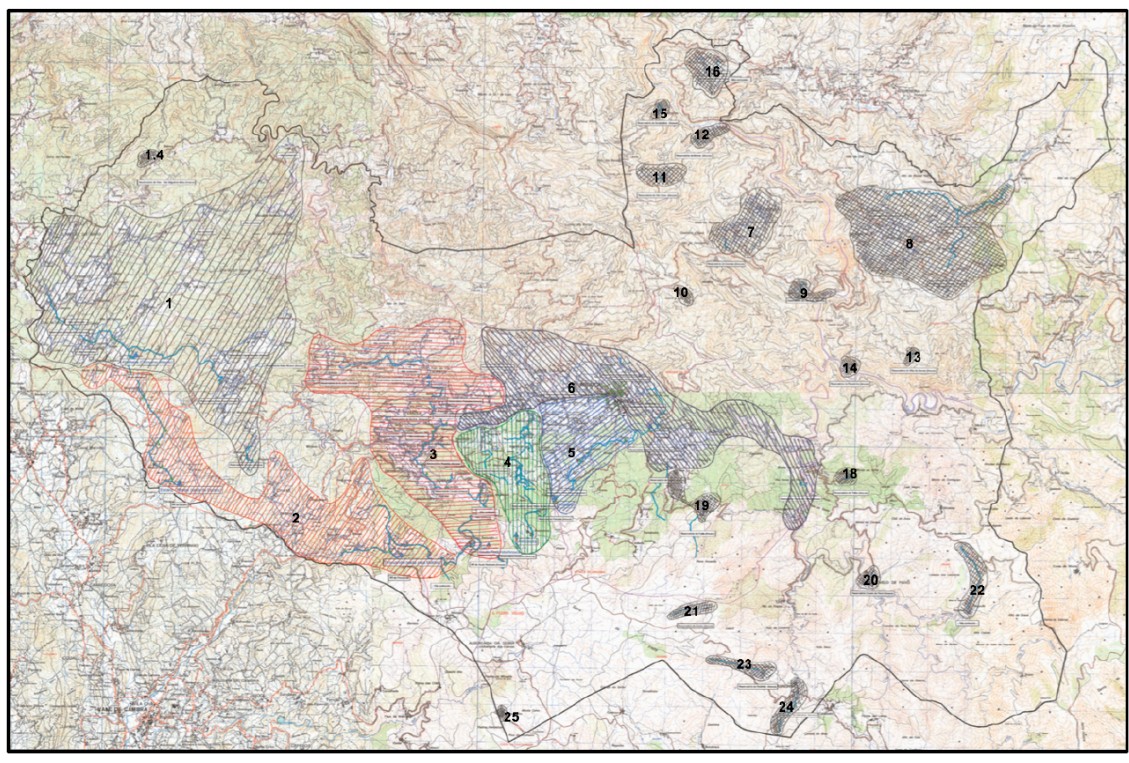

**Figure 5.** Supplied areas of the SAA-Arouca in 2017 (adapted from [23]).

Currently, the SAA-Arouca has 40 water catchments (13 out of service), 52 reservoirs (9 out of service), 27 pumping stations (7 out of service), 44 km of main pipelines, 17 water treatment plants, and 370 km of distributor pipelines.

In 2017, Arouca registered 10,695 houses, from which 7660 had available water supply (a coverage ratio of 72%), and from this portion, 6039 had effective water supply service (a subscribing ratio of 79%).

About 86% of the clients are domestic, followed by commercial/industrial/services with 8%. There was a pattern of evolution throughout the last 3 years, due to the number of clients, the execution of distributors, as well as substituting and installing water meters (thus ensuring better monitoring of the water volumes).

On this note, it is important to highlight that the age of the water meters may compromise the monitoring process of water volumes. Between 2016 and 2017, there was an improvement in the water meter fleet regarding the age of the equipment. This may have influenced, not solely, the improvements registered in Figures 6 and 7.

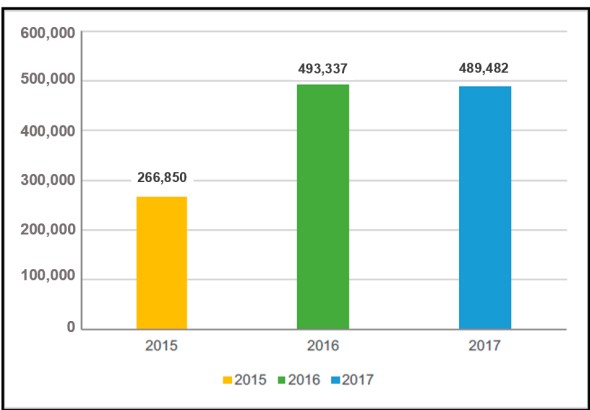

**Figure 6.** Volumes of water delivered in SAA-Arouca between 2015 and 2017 (adapted from [23]).

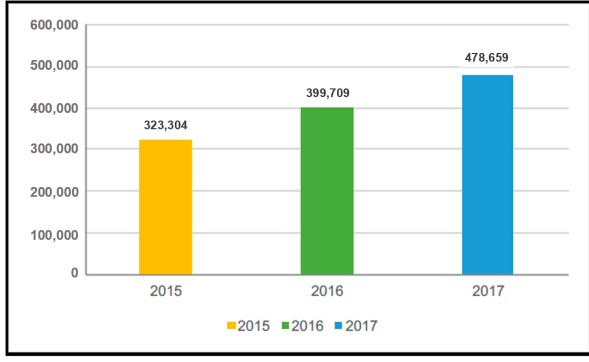

**Figure 7.** Volumes of water delivered in the MS between 2015 and 2017 (adapted from [23]).

Figures 6 and 7 reveal a difference between the volumes of water delivered in the SAA-Arouca globally and in the Main System only. In the latest, the volumes for the three years under observation are higher, and there is an increase in this indicator throughout the three years, while in the global analysis, there is an increase, though slight, between 2016 and 2017.

*3.2. Wastewater Drainage System of Trofa*

The Wastewater Drainage System of Trofa (SAR-Trofa, in Portuguese) drains the municipality of Trofa, which covers an area of 71.71 km$^2$ and with approximately 400,000 inhabitants. Regarding the year 2017, Trofa registered 15,583 houses. Of the 15,121 wastewater drainage options available (a ratio of 97% coverage), 9873 had effective service (a subscription ratio of 65%).

The Wastewater Drainage System of Trofa (SAR-Trofa, in Portuguese) is subdivided into two levels: the lower level, with 2 drainage basins, managed by the Northwest Region's Water System (SARN, in Portuguese), and the higher level, which is managed by TRATAVE (Tratamento de Águas Residuais do Ave, S.A.), responsible for SIDVA (translated to Depollution Integrated System of Ave's Valley).

SIDVA was developed to respond to the deterioration of water quality in the River Base of Ave's Valley (which has been aggravated since the 1980s). This system (SIDVA) encompasses four drainage fronts—Lordelo, Serzedelo, Rabada, and Agra. SIDVA is substantially exposed to groundwater and rainwater inflows because a substantial extension of the system is installed in the riverbed.

Regarding the SAR-Trofa organization, at the higher level of the system there are four interceptors—Trofa (managed by TRATAVE), Trofa Prolongamento, Covelas Poente e Bougado (all three managed by Águas do Norte, S.A.). At the lower level, there are drainage basins managed by Águas do norte, S.A., which connect to Trofa Interceptor. This Trofa Interceptor is then connected to the Ave Interceptor, which is included in the Multi-municipal System of Water Supply and Wastewater Drainage of Ave's Valley. There are a total of 11 drainage basins in the SAR-Trofa. Considering other infrastructure, there are 3925 connection branches, 99,251 m of collectors, interceptors, and outfalls, 2827 manholes, 3928 junction chambers, and five lifting installations. Figure 8 illustrates the organization of SAR-Trofa.

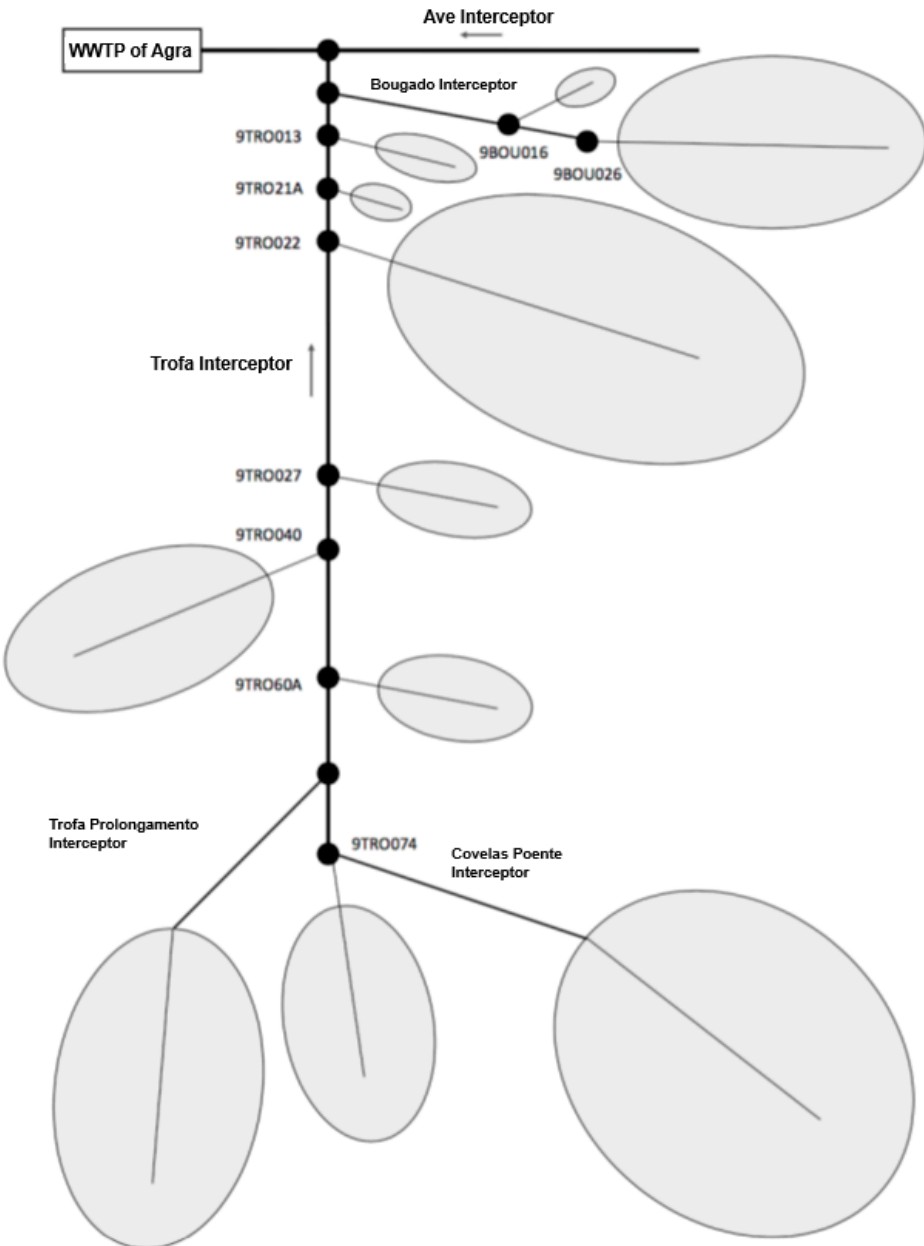

**Figure 8.** Schematic representation of the SAR-Trofa (adapted from [23]).

Service-wise, the coverage ratio has risen from 12% to 97% (year 2018) in the last 15 years; the number of clients rose between 2006 and 2018 to 6468, from a total of 4752 associated clients to 11,220—an approximately 58% increase.

The significant increase noted in Figure 9 between 2017 and 2018 was due to the installation of a flow meter in the drainage basin 9TRO022, making up a total of 2 flow meters in all of SAR-Trofa.

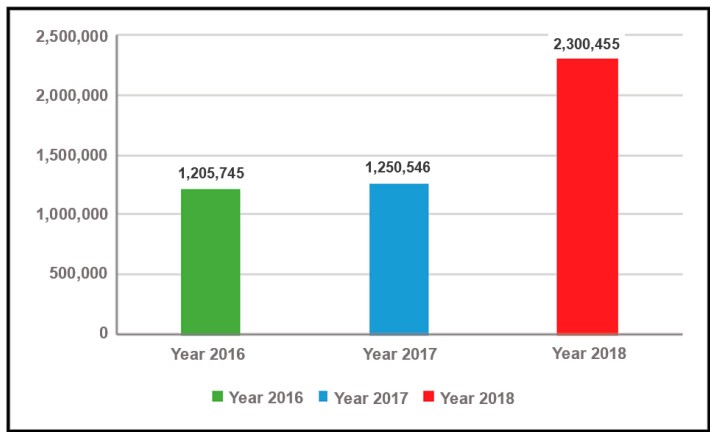

**Figure 9.** Volumes of delivered wastewater annually at the higher level by Águas do Norte, S.A., in Trofa, between 2016 and 2018 (adapted from [23]).

In 2018 (Figure 10), between 15% and 24% of the accounted volumes registered of TRATAVE were from industrial sources, which means that the volumes accounted are mainly domestic—850,032 m$^3$.

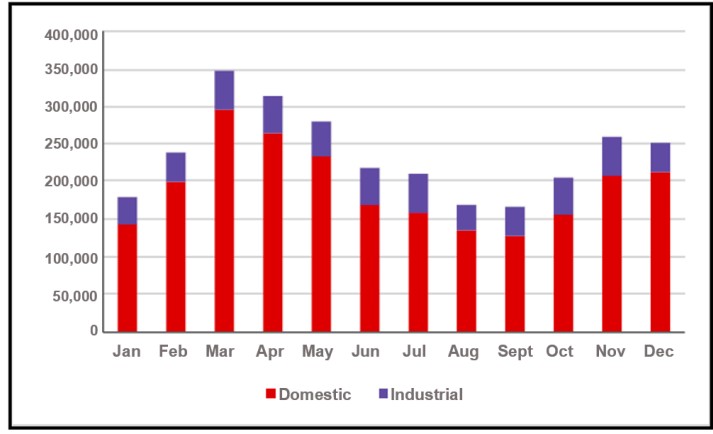

**Figure 10.** Graphic representation of monthly values accounted for by TRATAVE in Trofa in 2018 (adapted from [23]).

According to a study from Engidro e HIDRA (2006c), monitoring the Trofa interceptor on inflows showed this was the worst-performing interceptor (out of the four aforementioned) at the higher level. This interceptor showed that the flow from upstream to downstream nearly doubled (58 m$^3$/h and 111 m$^3$/h).

## 4. Application of the Methodology GOIH

### 4.1. SAA-Arouca

#### 4.1.1. Assessment

Infrastructure status assessment is done at five levels—very bad, bad, fair, good, and very good. In 2018, the global status of SAA-Arouca was as follows:

Catchments—87% are in fair or bad status; 13% are in good status.

Pipelines and distributors:

Pipelines—mainly in fair or very good status (31% and 25%, respectively).

Distributors—mainly in fair status (77%).

Reservoirs (construction and equipment)—mainly in fair status (47% and 55%, respectively).

Treatment System—70% in fair status (in construction and equipment).

Pressure Reducing Valve—regarding construction, it is mainly in fair or good status (42% and 24%); regarding equipment, it is mainly in good status (71%).

Pumping Stations—77% in fair status (construction and equipment).

Regarding performance assessment, the same infrastructures in the Status Assessment, were established at five levels—ineffectiveness, pre-effectiveness, effectiveness, efficiency, and excellency. The results of the assessments were:

Catchments—regarding flow availability, between pre-effectiveness, effectiveness, and efficiency, with 26%, 43%, and 30%, respectively.

Pipelines and distributors:

Pipelines—functioning wise, they present a similar pattern to the Catchments, with 28% for pre-effectiveness, 47% for effectiveness, and 25% for efficiency.

Distributors—regarding functioning, these infrastructures are divided into pre-effectiveness (40%), effectiveness (24%), and efficiency (35%); as for accessory suitability, the registered status was mainly pre-effectiveness (50%) and effectiveness (46%).

Reservoirs—assessing reservation, level control, and watertightness, the main status level of each was effectiveness (40%), pre-effectiveness (68%), and pre-effectiveness (40%), respectively.

Treatment System—these were assessed regarding treatment capacity and functioning. Treatment capacity was assessed mostly by the level of effectiveness (70%). As for functioning, a more balanced assessment was registered between pre-effectiveness, effectiveness, and efficiency—40%, 24%, and 35%, respectively.

Pressure Reducing Valve—regarding pressure reduction, these devices were assessed with mainly efficiency (90%); as for flow availability, pre-effectiveness and effectiveness were balanced, with 42% and 45%, respectively.

Pumping Stations—these were assessed considering pumping capacity (41% effectiveness and 55% efficiency) and energetic efficiency (32% pre-effectiveness and 68% effectiveness).

Social Performance was also assessed, and, with one exception, all the indicators were predominantly effective or efficient when compared with the infrastructure assessment:

Image of the Management Entity—only 5% have a pre-effectiveness level; the remaining 95% are divided into effectiveness (37%), efficiency (53%), and excellency (5%).

Customer Service—like the above-mentioned indicator, 5% was attributed to a pre-effectiveness level; the remaining was subdivided into 42% effectiveness, 32% efficiency, and 21% excellency.

Invoicing Process—this indicator was the one exception that showed a higher pre-effectiveness level (47%), with the remaining divided among the three highest levels of the attribution scale.

Service Quality—regarding this, effectiveness and efficiency were predominantly registered, with values of 42% and 47%, respectively.

Occurrence Response—predominantly efficient, with 63% and 16% at the excellency level.

Problem Resolution—52% of effectiveness was assessed, and 32% efficiency.

Meters Installment—16% was found at the excellency level; the remaining was equally divided between effectiveness and efficiency.

Execution of Branches—mainly divided into effectiveness (42%), efficiency (37%), excellency (16%), and pre-effectiveness (5%).

Infrastructure Conservation Status—mostly assessed with the efficiency level (43%), followed by 32% with effectiveness, 11% with pre-effectiveness, and 5% with excellency.

Infrastructure Safety—42% of the system is at an efficiency level, and 37% is at an effectiveness level.

Global Assessment—globally, this system's social performance is efficient in 43%, effective in 37%, excellent in 12%, and pre-effective in 8%.

Cost assessment is summarized in the following tables (Tables 1 and 2), regarding infrastructure consumed, useful life, current value, and replacement cost:

**Table 1.** Infrastructure Cost Assessment of SAA-Arouca (2017)—Summary (adapted from [23]).

| Infrastructure | Replacement Cost | Current Value | Infrastructure Value Index |
|---|---|---|---|
| Catchments | EUR 158,625 | EUR 99,460 | 0.63 |
| Pipelines | EUR 1,381,695 | EUR 901,544 | 0.65 |
| Reservoirs | EUR 2,353,779 | EUR 1,425,292 | 0.61 |
| Treatment System | EUR 60,800 | EUR 35,760 | 0.59 |
| Distributors | EUR 15,200,880 | EUR 9,287,480 | 0.61 |
| Pressure Reducing Valve | EUR 590,614 | EUR 409,441 | 0.69 |
| Pumping Station | EUR 99,000 | EUR 61,100 | 0.62 |
| **Total** | **EUR19,845,393** | **EUR12,220,077** | **0.62** |

**Table 2.** Infrastructures Rehabilitation Estimated Cost of SAA-Arouca (2017)—Summary (adapted from [23]).

| Infrastructure | Consumed Useful Life |
|---|---|
| Catchments | 37% |
| Pipelines | 35% |
| Reservoirs | 39% |
| Treatment System | 41% |
| Distributors | 39% |
| Pressure Reducing Valve | 31% |
| Pumping Station | 38% |
| **SAA-Arouca (Global)** | **38%** |

This infrastructure value index (IVI) of around 60% indicates that infrastructure is stable. Risk was assessed based on a 5-level scale—very low, low, average, high, and very high:

Catchments—non-compliance is mostly at a high or average level of risk (43% and 48% respectively). Regarding safety, is between low risk (70%) and average (30%).

Pipelines and distributors:

Pipelines—risk was assessed based on the probability of water loss, rupture, and service interruption. For water loss and service interruption, the probability was mainly average (53% and 54%, respectively). Regarding rupture risk, the probability lies at a low level, at 45%.

Distributors—the same as abovementioned was assessed in distributors. Regarding the risk probability for water loss and rupture, it was registered at 39% and 38% of the average risk, respectively, followed by 45% and 42% at a high level of risk. As for service interruption, 54% were at the average level and 43% at the low level of risk.

Reservoirs—were assessed for water loss, contamination, and accidents. For these 3, the average level of risk represented 40% to 43%, followed by a high level of risk between 34% and 40%.

Treatment System—service interruption was at the base of the assessment, with 82% in average risk.

Pressure Reducing Valve—the parameters used to assess these devices have all been above 88% of low risk—water loss (93%), rupture (88%), and service interruption (92%).

Pumping Stations—the parameters used were water loss (59% in low risk level), contamination (41% in low risk level and 50% in average risk level), and accident (32% in low risk level and 54% in average risk level).

In the following table (Table 3) is a summary of the several assessments aforementioned according to SAA-Arouca's subsystems.

**Table 3.** Infrastructure Assessment of SAA-Arouca—Summary (adapted from [23]).

| Assessment | Abelheira | Provisende | S. Redondo | Ameixieira | Forcada | Moldes | Autonomous Systems | Average |
|---|---|---|---|---|---|---|---|---|
| **Status** | 2.49 (good) | 2.45 (good) | 2.64 (fair) | 3.42 (fair) | 2.82 (fair) | 3.11 (fair) | 3.36 (fair) | 2.88 (fair) |
| **Performance** | 2.20 (efficiency) | 2.32 (efficiency) | 2.70 (efficacy) | 3.30 (efficacy) | 3.27 (efficacy) | 3.07 (efficacy) | 3.30 (efficacy) | 2.86 (efficacy) |
| **Risk** | 6.07 (intermediate) | 4.38 (low) | 8.02 (intermediate) | 7.81 (intermediate) | 6.22 (intermediate) | 8.13 (intermediate) | 8.69 (intermediate) | 7.77 (intermediate) |
| **Actions/interventions to consider** | Pressure control | Sectioning valves and Landfills discharges | Pipeline replacement in São João reservoir | Pressure control and pipeline replacement in Fráguas reservoir | Connection execution from Forcada reservoir to Santo Aleixo | Pressure control and replacement of part of the supply network | Pressure control, air valve installation, mine inspection, and deviation of part of the catchment's water | |

### 4.1.2. Exploration

Concerning operational control, there is a summary of the proposed and implemented actions with the application of GOIH methodology. Several actions were proposed:

Reservoir monitoring.

Systematic registration and management of occurrences.

Structured management of various works.

Monitoring standpipes with autonomous sources.

Setting up and managing a Pressure Reducing Valves team.

Operational costs were slightly lower in 2018 when compared with the previous year (Figure 11).

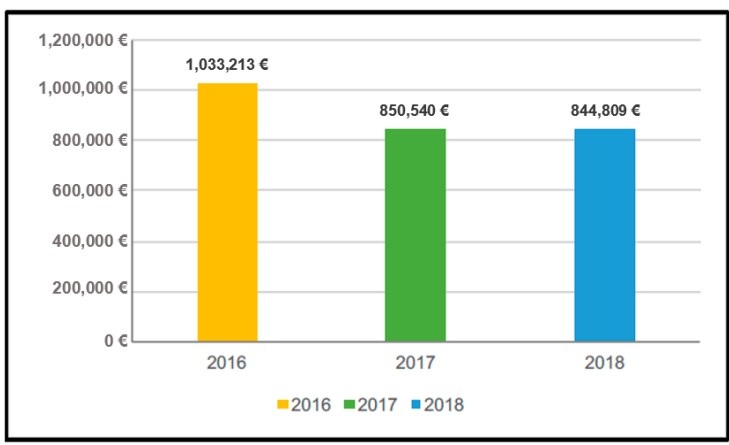

**Figure 11.** Graphic representation of the annual operational costs of SAA-Arouca between 2016 and 2018 (adapted from [23]).

Regarding human resources management, the proposed functional teams were designed to account for the main indispensable routines for adequate hydraulic infrastructure management:

System monitoring:

Water quality.

Volumes and consumption.

Energy consumption.

Verification of electromechanical equipment.

Prevention:

Response to occurrences.

Resolution of urgent problems.

Losses:

Operational optimization.

Active leakage control.

Night and day inspections.

Pressure Reducing Valves:

Monitoring, maintenance, replacement, and installation of.

Pressure management.

Ruptures:

Identification.

Repairment.

Monitoring of work carried out by external providers.

Reduction of response times.

The proposed actions in response to occurrences have the goal of promoting a timely and effective response to the different demands of managing SAA-Arouca. The proposal is as follows:

Reduction of response time through:

Optimizing the process of sending internal teams to identify incidents.

Systematic monitoring and control of external providers.

Management of the SAA-Arouca, accounting for the relevance of the subsystems, and enabling:

Continuity of service.

Data minimization.

The real data collected regarding the Main System shows an increase in water shortage, probably due to the supply network extension that happened in the network dependent on the delivery point of Souto Redondo (Figure 12), which can be associated with a lack of delivery capacity and ruptures on the pipeline of São João reservoir.

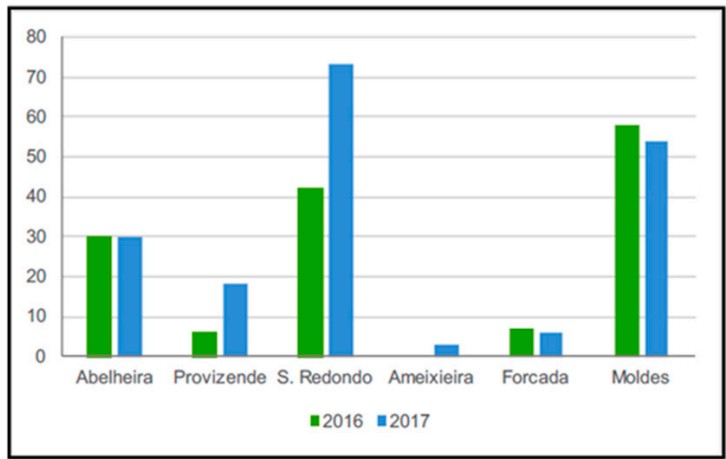

**Figure 12.** Graphic representation of water shortages in the Main System between 2016 and 2017 (adapted from [23]).

Water quality had non-conformities mainly in the Autonomous Systems (around 78% of the non-conformities), due to the treatment systems having less reliability when compared to the Main System.

In Figure 13, it is possible to note that a higher number of ruptures occurred between June and August, over 50% compared to the average monthly ruptures.

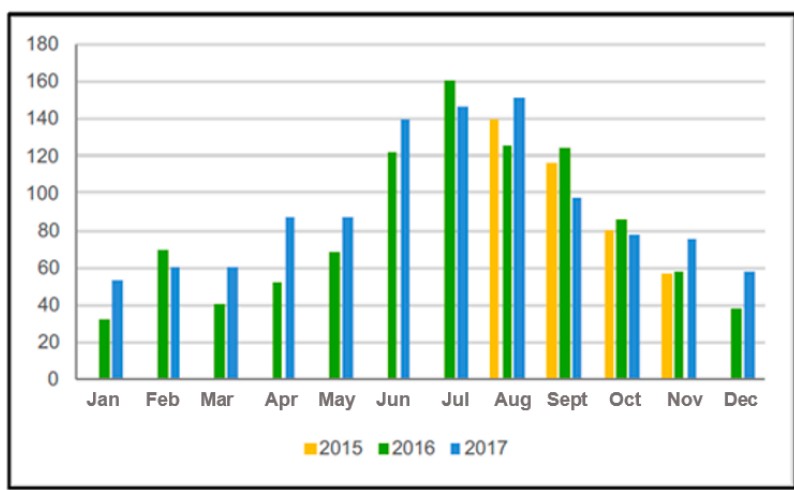

**Figure 13.** Graphic representation of ruptures that occur monthly in SAA-Arouca between 2015 and 2017 [23].

Risk management was translated by the rupture costs and real repair costs from 2015 to 2017. The ratio between the associated total costs and the number of ruptures that occurred allows the calculation of the average cost per rupture for each subsystem, being the subsystem of Moldes delivery point that presents the highest risk of rupture (Figure 14).

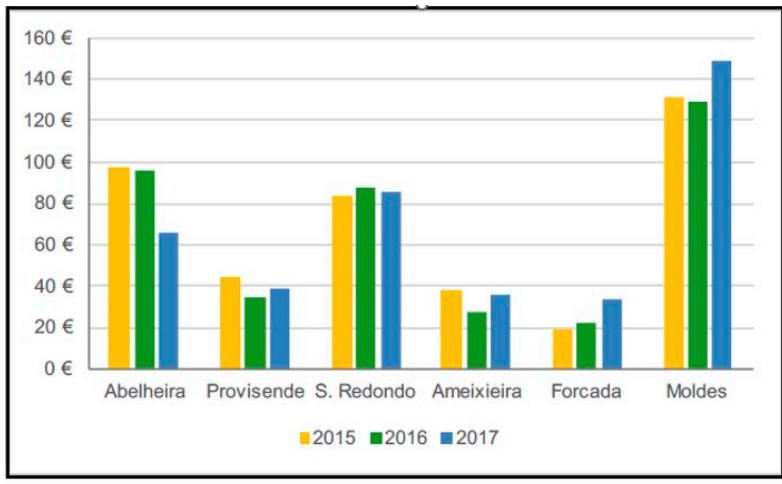

**Figure 14.** Annual rupture risk in the Main System by delivery point between 2015 and 2017 (adapted from [23]).

Figures 15–17 show the risks for ruptures, water shortages, and water loss, respectively, translated into associated costs. The subsystems of Moldes, Souto Redondo, and Abelheira are the ones with higher annual rupture risk (Figure 15), which can be translated into monetary values (Figure 15). There also appears to be a correlation between the rupture risk and water shortage, as Figure 16 shows a similar pattern to Figures 14 and 15. Figure 17 shows water loss translated into monetary values, and the same pattern seems to apply regarding these three more concerning subsystems.

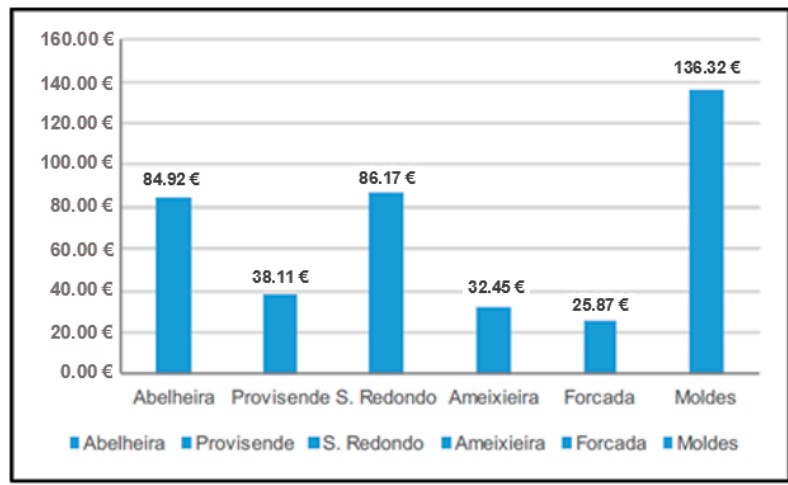

**Figure 15.** Annual rupture risk in the Main System between 2015 and 2017 (adapted from [23]).

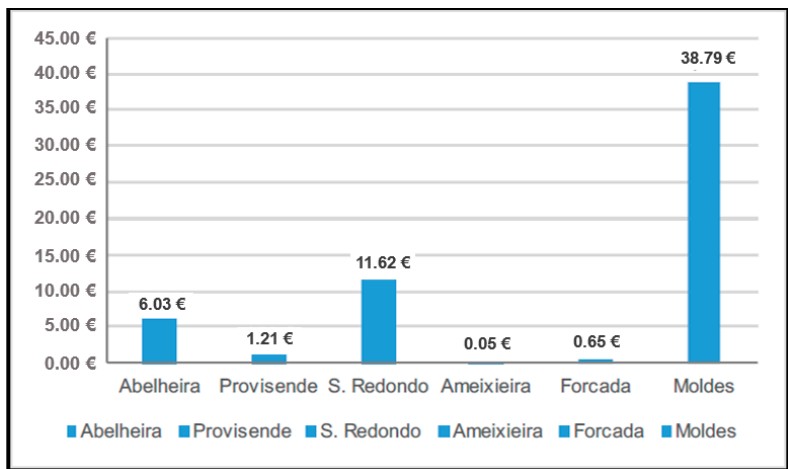

**Figure 16.** Risk of water shortage in the Main System between 2015 and 2017 (adapted from [23]).

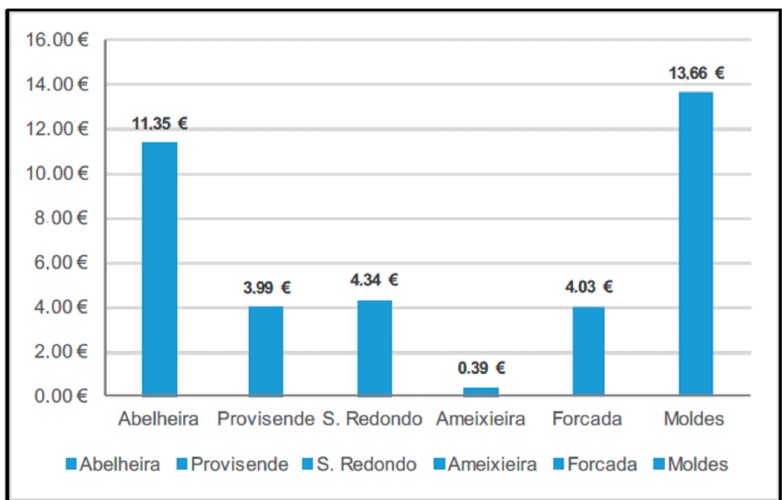

**Figure 17.** Water loss risk in the Main System between 2015 and 2017 (adapted from [23]).

In order to improve the problems of SAA-Arouca, several actions were proposed and developed.

The methodology ends up in the exploration section with several proposed actions for the future, focused on operational control and apparent losses, such as:

Reservoir monitoring and occurrence management with Aquafield;

Consumption monitoring;

Team planning and management;

Supplies management;

Reduction of losses due to measurement errors, not-accounted consumptions, and not-authorized consumptions.

The table that follows (Table 4) summarizes the costs of possible improvements for functional teams.

**Table 4.** Summary of means allocation and associated costs for functional teams of the SAA-Arouca (adapted from [23]).

| Functional Teams | Human Resources | | Vehicles | | Supplies | | Total Cost |
|---|---|---|---|---|---|---|---|
| | Allocation | Cost | Allocation | Cost | Allocation | Cost | |
| Prevention | 30% | EUR 47,520 | 30% | EUR 5850 | 60% | EUR 12,000 | EUR 65,370 |
| Monitoring SAA | 10% | EUR 15,840 | 30% | EUR 5850 | 5% | EUR 1000 | EUR 22,690 |
| Ruptures/Clearings | 10% | EUR 15,840 | 10% | EUR 1950 | 10% | EUR 2000 | EUR 19,790 |
| Pressure Reducing Valve | 20% | EUR 31,680 | 15% | EUR 2925 | 20% | EUR 4000 | EUR 38,605 |
| Loss/Infiltrations | 30% | EUR 47,520 | 15% | EUR 2925 | 5% | EUR 1000 | EUR 51,445 |
| **Total** | | **EUR 158,400** | | **EUR 19,500** | | **EUR 20,000** | **197,900** |

Actions already implemented focused on combating real water losses by outlining the water loss prevention zone (ZCP, in Portuguese). Seven ZCPs were outlined and intervened through: maintenance, replacement, installation, and adjustment of pressure-reducing valves; network replacement; rupture repairment. In Table 5, it is possible to see the improvement of six in the ZCP after the interventions.

**Table 5.** Summary of the results of active water loss control in the Main System in 2018 (adapted from [23]).

| ZCP | Period (2018) | Minimal Night Consumption | | | Annual Estimated Gains | |
|---|---|---|---|---|---|---|
| | | Initial | Final | Decrease | | |
| ZCP Pressure Reducing Valve | June—July | 7.0 m$^3$/h | 4.0 m$^3$/h | 4.0 m$^3$/h (− 57%) | 35,040 m$^3$ | EUR 18,245 |
| | November—December | 3.5 m$^3$/h | 2.5 m$^3$/h | | | |
| ZCP Vila Nova | September—October | 11.0 m$^3$/h | 3.0 m$^3$/h | 8.0 m$^3$/h (− 73%) | 70,080 m$^3$ | EUR 36,491 |
| ZCP São Pedro | November—December | 7.0 m$^3$/h | 1.0 m$^3$/h | 6.0 m$^3$/h (− 86%) | 52,560 m$^3$ | EUR 27,368 |
| ZCP Abelheira | November—? | * 20.0 m$^3$/h | * 18.0 m$^3$/h | 2.0 m$^3$/h (− 10%) | 17,520 m$^3$ | EUR 9123 |
| ZCP Provisende | November—? | 7.0 m$^3$/h | 6.0 m$^3$/h | 1.0 m$^3$/h (− 14%) | 8760 m$^3$ | EUR 4561 |
| ZCP Forcada | December—? | 9.0 m$^3$/h | 6.4 m$^3$/h | 2.6 m$^3$/h (− 29%) | 22,776 m$^3$ | EUR 11,860 |
| | **Total:** | | | **23.6 m$^3$/h** | **206,736 m$^3$** | **EUR 107,648** |

Note: * The values of de Minimal Night Consumption of ZCP Abelheira include the continuing supply to the Reservoir of Cimo da Inha, with an intake of approximately 10 m$^3$/h.

The ZCP Canelas are not included in Table 5, but they showed improvement after the intervention in regard to pressure control, minimal night consumption, and rupture number. Figures 18 and 19 show this improvement. Actions to prevent water losses were implemented in the second semester of 2018, and these figures show a clear decrease in minimal night consumption (Figure 18) and a decrease in rupture number (Figure 19).

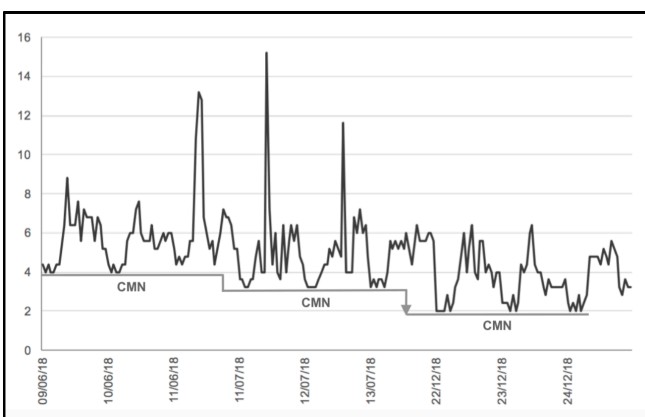

**Figure 18.** Minimal Night Consumption (CMN, in Portuguese) Evolution in ZCP Canelas (adapted from [23]).

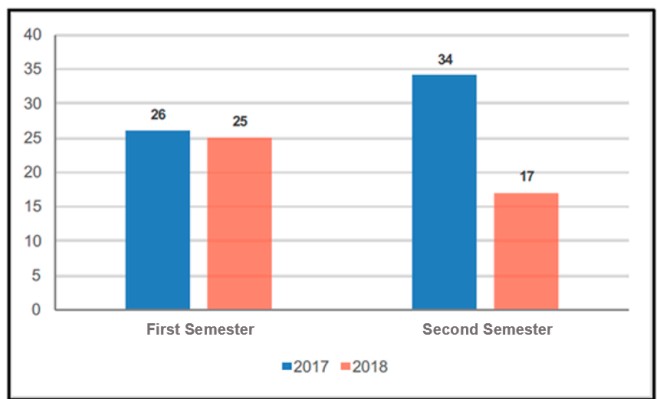

**Figure 19.** Rupture number evolution in ZCP Canelas (adapted from [23]).

4.1.3. Intervention

Regarding *Monitoring* interventions, they were focused on pressure-reducing valves (VRP, in Portuguese), reservoirs, water fountains with uncontrolled sources, and a systematic record of occurrences. VRP suffered pressure management, as shown on the following table (Table 6), and it is possible to note a total reduction of 18% downstream of the VRP, therefore pressure on the supply system overall. It is important to mention that the pressure reduction was done without detriment to the continuity and quality of the service.

**Table 6.** Summary of Pressure Management in SAA-Arouca (adapted from [23]).

| Subsystem | VRP Number | VRP with Pressure Reduction | Pressure Reduction after Fase 1 on Pressure Management | |
|---|---|---|---|---|
| | | | (mca) | Final |
| Abelheira | 33 | 17 | 8.5 | 26% |
| Provisende | 14 | 10 | 7.0 | 23% |
| S. Redondo | 40 | 12 | 4.2 | 13% |
| Ameixieira | 11 | 4 | 5.7 | 18% |
| Forcada | 15 | 7 | 4.4 | 12% |
| Moldes | 30 | 11 | 6.7 | 19% |
| **Total** | **143** | **62** | **6.0** | **18%** |

*Maintenance* was focused on VRP and reservoirs, emphasizing the Reservoir of Vista Alegre and the Reservoir of Cimo da Inha.

As for *Rehabilitation*, portions of the network and VRP were replaced with the costs presented in Table 7.

**Table 7.** Summary of Rehabilitation and Reinforcement actions (adapted from [23]).

| Subsystems | Replacement Length | Replacement Cost | VRP Replacement Cost | Installation Length | Installation Cost | VRP Installation Cost |
|---|---|---|---|---|---|---|
| Abelheira | 1660 m | EUR 24,827 | | 1795 m | EUR 69,804 | |
| Provisende | 250 m | EUR 6520 | | 285 m | EUR 3111 | |
| S. Redondo | 4014 m | EUR 88,927 | EUR 21,000 *10 VRP were replaced* | 2926 m | EUR 73,057 | EUR 81,000 *3 VRP were installed* |
| Ameixieira | 40 m | EUR 1600 | | 70 m | EUR 2800 | |
| Forcada | 120 m | EUR 4800 | | 386 m | EUR 6514 | |
| Moldes | 870 m | EUR 16,931 | | 310 m | EUR 12,004 | |
| Autonomous Systems | 615 m | EUR 8370 | | 300 m | EUR 5474 | |

As for *Reinforcement* measures, conducts and VRP were installed, as well as level controls in reservoirs and maneuvering chambers (costs presented in Table 7).

Interventions are expected to continue as progressive work that has already started, as follows (Table 8):

**Table 8.** Summary of the interventions carried out in SAA-Arouca between 2016 and 2017 (adapted from [23]).

| Year | Rehabilitation | | Reinforcement | | Total |
|---|---|---|---|---|---|
| | Conducts | VRP | Conducts | VRP | |
| 2016 | EUR 83,235 | - | - | - | EUR 83,235 |
| 2017 | EUR 202,425 | EUR 21,000 | EUR 172,764 | EUR 81,000 | EUR 477,189 |
| 2018 | EUR 101,663 | EUR 25,250 | EUR 109,597 | EUR 57,500 | EUR 294,010 |
| **Following** | EUR 463,553 | EUR 52,500 | EUR 1,522,501 | EUR 123,000 | EUR 2,161,554 |
| **Total** | **EUR 850,876** | **EUR 98,750** | **EUR 1,804,862** | **EUR 261,500** | **EUR 3,015,988** |

### 4.1.4. Overview

The following figures show graphically the improvement of the Main System, which is also reflected in the SAA-Arouca, regarding rupture occurrences (Figure 20), system input volumes (Figure 21), billed consumptions, and non-revenue water (Figure 22). It is a clear improvement and successful implementation of the GOIH methodology, with important results for the efficiency and sustainability of this water supply system, as all of these indicators have evolved desirably.

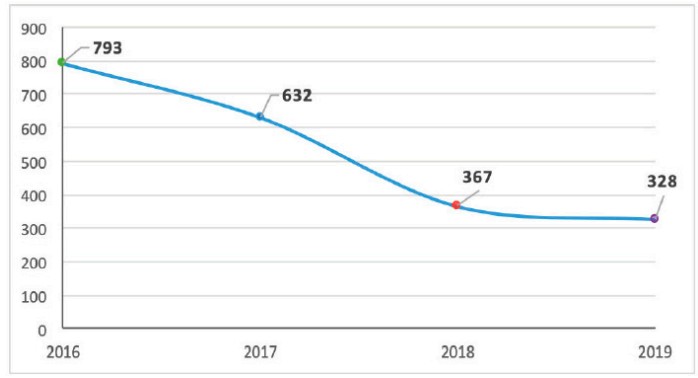

**Figure 20.** Evolution of the number of ruptures in the Main System between 2016 and 2019 (adapted from [2]).

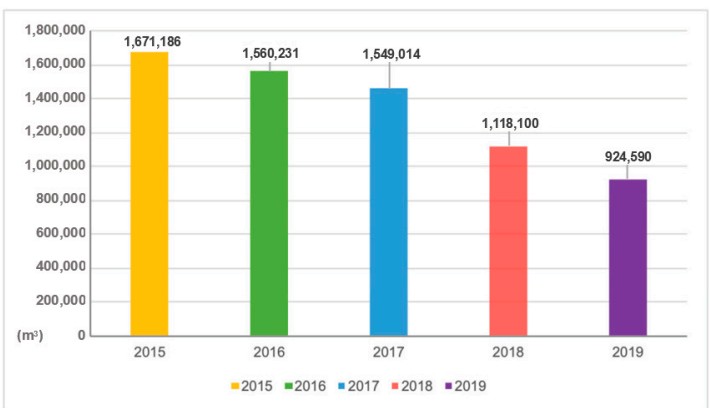

**Figure 21.** System Input Volumes the Main System of SAA-Arouca between 2015 and 2019 (adapted from [2]).

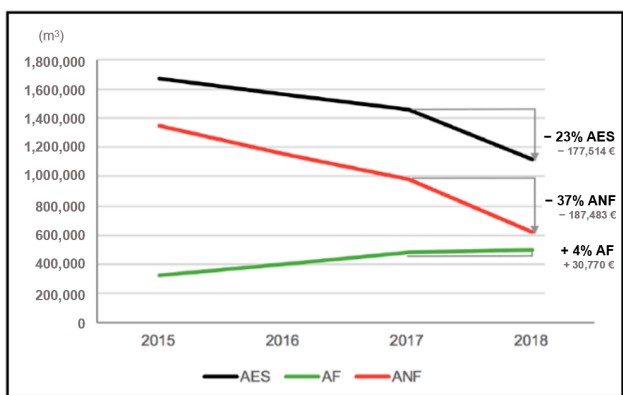

**Figure 22.** Evolution of water entering the system (AES), accounted water (AF), and not accounted water (ANF) in the Main System between 2015 and 2018 (adapted from [23]).

4.1.5. Decision Support

Nine investment scenarios were calculated (from no investment at all to an investment of EUR 626,020 with a return period of 20 years) in this analysis. Considering that accounted water remains constant, in Figure 23, it is possible to analyze three possible scenarios, from a more conservative one to the least conservative:

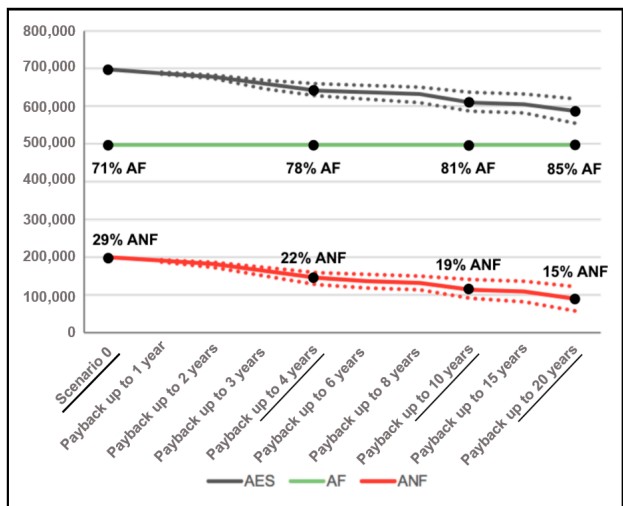

**Figure 23.** Prediction of the evolution of water entering the system, accounted water and not accounted water (adapted from [23]).

It is important to note that all the scenarios in Figure 23 ensure the quality of this service. The suggested approximate investments with operational costs were as follows:

Investments for 2019:

Inventoried interventions—EUR 191,000.

Interventions in execution—EUR 94,000.

Operational teams—EUR 200,000.

Total investment—EUR 485,000.

Investments for the following years:

Inventoried interventions—EUR 300,000.

Operational teams—EUR 200,000.

Total investment—EUR 500,000.

Investments with a strategic return:

Improvement of the Main System—EUR 544,000.

Improvement of the Autonomous Systems—EUR 658,000.

Total investment—EUR 1,200,000.

The continued application of GOIH allowed for the continued improvement of SAA-Arouca (Figure 24), showing a 64% reduction in water input between 2015 and 2023.

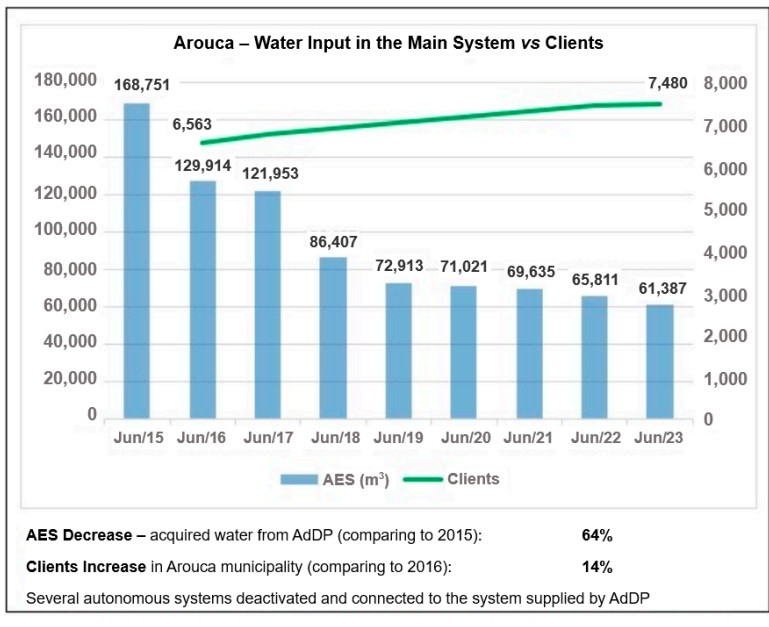

**Figure 24.** Evolution of water input in June between 2015 and 2023.

*4.2. SAR-Trofa*

4.2.1. Assessment

The status assessment registered that most of the connection lines were in good status (83%), 5% of them were in fair status, and the remaining 12% were in bad status. As for collectors and interceptors, 59% were in fair status, 35% were in good status, and finally, 6% were in bad status. In neither infrastructure were registered very good or very bad status.

Finally, manholes were, for the most part, in bad status (42%), 23% were in fair status, and the remaining 32% were in good status.

The performance assessment was done using a five-level scale (from worst to best—inefficacy, pre-efficacy, efficacy, efficiency, and excellency) and showed that all the indicators are within the levels of pre-efficacy and efficiency.

Connection Lines—regarding operation and suitability. A total of 66% of these infrastructures were at an efficacy level, and 34% were efficient.

Collectors and Interceptors—as for operation assessment, 18% were in pre-efficacy, 58% were in an efficacy level, and 24% were in an efficiency level. About accessory suitability, 17% were in pre-efficacy, 30% were in efficacy level, and the remaining 53% were efficient.

Manholes—these infrastructures were mainly at an efficacy level with 57% of the infrastructures; as for accessibility, 55% were efficient.

Lift Installation—these were assessed on lifting capacity and energy efficiency. As for lifting capacity, all the infrastructures were efficient (100%); on energy efficiency, 80% were efficient.

The cost assessment was based on the infrastructure's useful and residual life, replacement cost, current value, and the infrastructure value index, summarized in Table 9.

**Table 9.** Infrastructure Cost Assessment of SAR-Trofa (adapted from [23]).

| Infrastructure | Replacement Cost | Current Value | Infrastructure Value Index |
|---|---|---|---|
| Connection Lines | EUR 834,064 | EUR 619,624 | 0.74 |
| Collectors and Interceptors | EUR 4,545,949 | EUR 2,952,769 | 0.65 |
| Manholes | EUR 1,413,500 | EUR 819,100 | 0.58 |
| Lift Installations | EUR 272,288 | EUR 206,384 | 0.76 |
| **Total** | **EUR 7,065,802** | **EUR 4,597,877** | **0.65** |

Risk assessment was based on a 5-level scale—very high, high, fair, low, and very low.

Connection Lines—these were assessed on obstruction probability and inflow probability. As for obstruction, 67% were at a fair level of risk, and 33% were at a low risk level. About inflows, 36% are at a very high-risk level, followed by 41% at a high risk.

Collectors and Interceptors—in 58% of these infrastructures were at a fair risk level of obstruction, 27% at a low risk. As for inflows, 27% were at a very high-risk level, and 61% were at a high-risk level. Also, 44% were at a fair risk of collapse, and 41% were at a low risk.

Manholes—58% of the manholes were at a fair risk level of obstruction, 27% were at a low risk level, and the remaining 15% were at a high level of risk. As for inflows, 26% were at a very high-risk level, and 60% were at a high level. Finally, accidents related to the environment and safety probability were also assessed, with 47% at a fair risk and 27% at a high risk.

Lift Installation—regarding these infrastructures, the probability of service interruptions, as well as inflows and accidents related to the environment and safety. Service interruption presents a fair risk level in 60% of these infrastructures, with the remaining 40% presenting a low risk. There is a high risk of inflows in 20% of the assessed infrastructure, a fair risk in 40%, and a low risk in the remaining 40%. As for the accidents' indicator, 80% presented a fair risk level, and the remaining 20% presented a low risk.

### 4.2.2. Exploration

Regarding the *Operational control* stage of this section, the main scheduled operational routines are the systematic record of occurrences, monitoring of lifting installations, and checking of critical points of the systems.

As for *Occurrence response*, the sites with significant relevance can be indicated using the operational relevance index (IRO, in Portuguese), based on the network extension, number of clients, and system input volumes (VEA, in Portuguese). This is shown in Table 10 and Figure 25 for each drainage basin.

**Table 10.** Operational Relevance Index in the drainage basins of SAR-Trofa (adapted from [23]).

| Drainage Basin | Operational Relevance Index (IRO) | | | | | | |
| --- | --- | --- | --- | --- | --- | --- | --- |
| | Extension (km) | | Clients (un) | | VEA (m³) | | Global |
| BD 9BOU016 | 1734 | 0.02 | 219 | 0.05 | 6363 | 0.00 | **0.02** |
| BD BOU026 | 14,850 | 0.15 | 524 | 0.13 | 398,801 | 0.24 | **0.17** |
| BD 9TRO013 | 1379 | 0.01 | 12 | 0.00 | 2741 | 0.00 | **0.01** |
| BD 9TRO21A | 428 | 0.00 | 6 | 0.00 | 623 | 0.00 | **0.00** |
| BD 9TRO022 | 26,891 | 0.28 | 2219 | 0.53 | 1,169,647 | 0.71 | **0.51** |
| BD 9TRO027 | 1108 | 0.01 | 20 | 0.00 | 3979 | 0.00 | **0.01** |
| BD 9TRO040 | 5385 | 0.06 | 104 | 0.02 | 12,918 | 0.01 | **0.03** |
| BD 9TRO60A | 1209 | 0.01 | 24 | 0.01 | 3793 | 0.00 | **0.01** |
| BD Trofa Prolongamento | 13,120 | 0.14 | 449 | 0.11 | 20,698 | 0.01 | **0.09** |
| BD 9TRO074 | 5748 | 0.06 | 270 | 0.06 | *12,151 ** | 0.01 | **0.04** |
| BD Covelas Poente | 24,059 | 0.25 | 344 | 0.08 | *15,482 ** | 0.01 | **0.11** |
| | **95,911** | | **4191** | | **1,647,196** | | |

Note: * Accounted volume (VEA) in these two delivery points are calculation jointly. The division was done with the proportion of the number of clients.

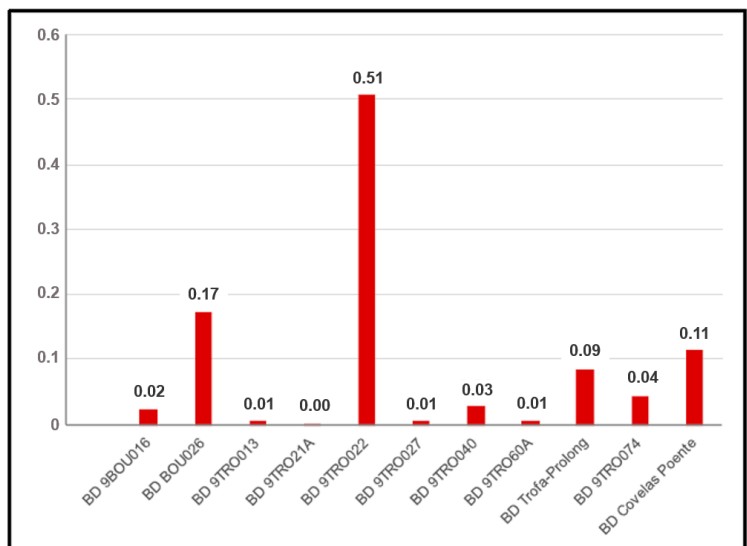

**Figure 25.** Graphic representation of the Operational Relevance Index of SAR-Trofa (adapted from [23]).

According to this, the drainage basin with the most operational relevance is BDTRO022, followed by BD BOU026, with a significant difference between the two, also notable in Figure 25.

As for *Information: Real Data*, energy consumption, obstruction, accounted volumes at high levels, and measured volumes.

In Figure 26, it is possible to note that the highest energy consumption is in BD TRO022. Regarding obstruction registered, Figure 27 shows no patterns in obstruction occurrences throughout the year or between 2017 and 2018.

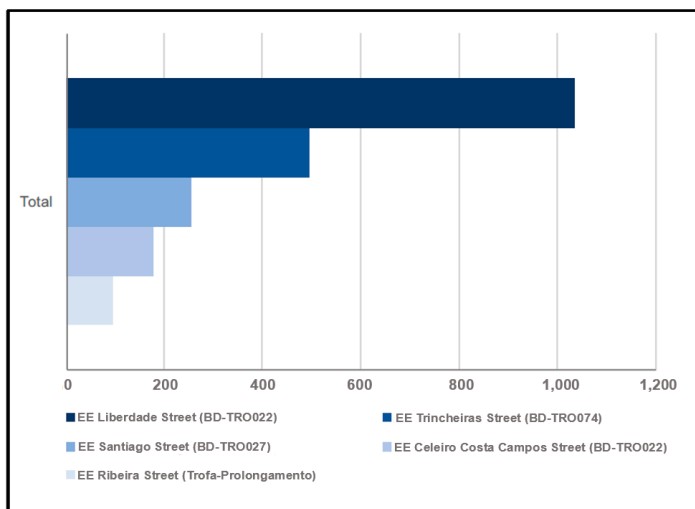

**Figure 26.** Annual Energetic Consumption (kWh) of Lift Installations of SAR-Trofa (adapted from [23]).

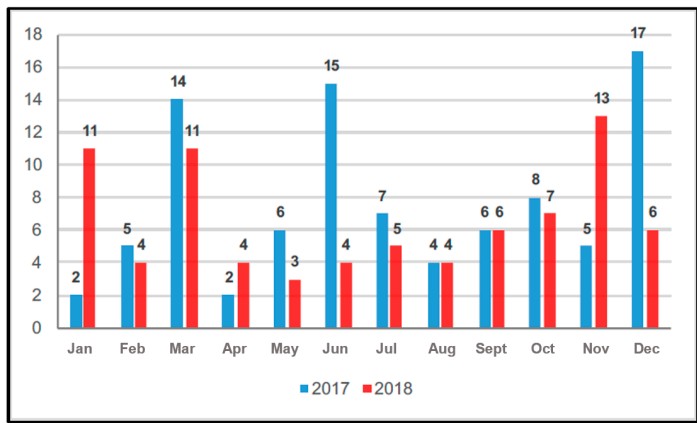

**Figure 27.** Monthly distribution of Obstructions occurrence in Trofa municipality between 2017 and 2018 (adapted from [23]).

As for volumes registered, the following figures (Figures 28–30) show a bigger similarity between Figures 28 and 30, which can indicate that basin BD 9TRO022 influences the whole system significantly.

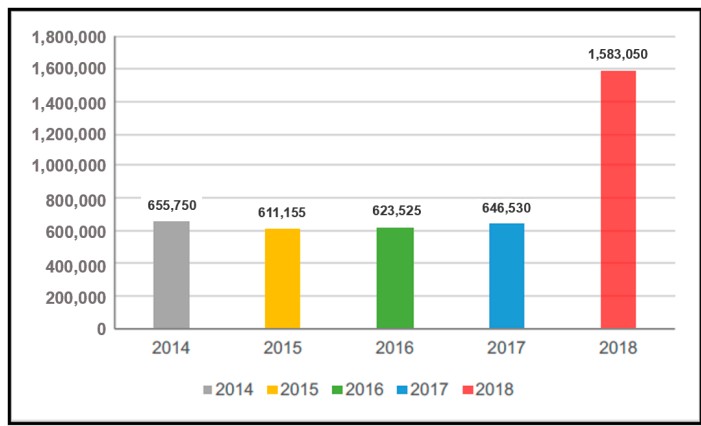

**Figure 28.** Quantified volumes in the drainage basins from 2014 to 2018 (adapted from [23]).

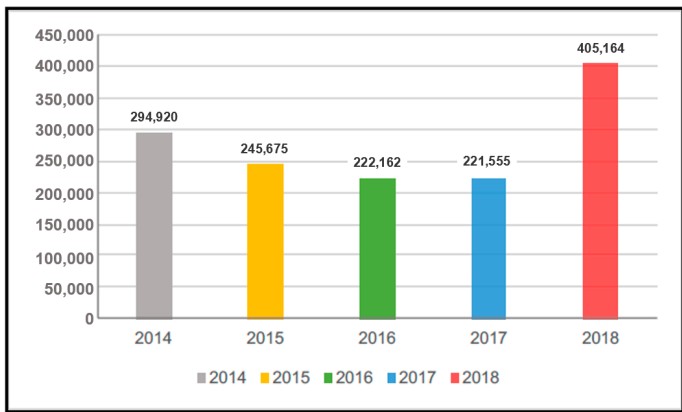

**Figure 29.** Quantified volumes in the drainage basin of the Interceptor of Bougado between 2014 and 2018 (adapted from [23]).

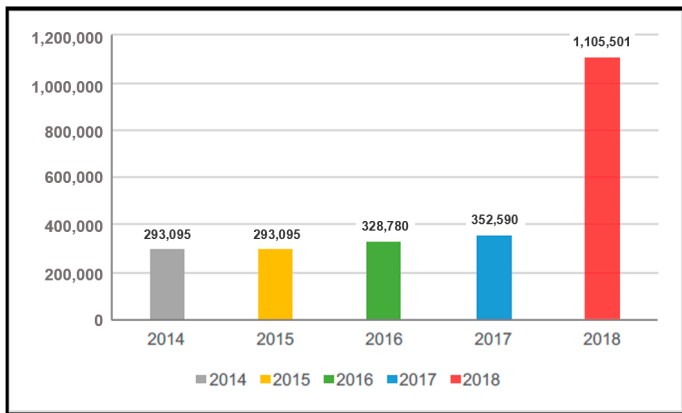

**Figure 30.** Quantified volumes in the drainage basin BD 9TRO022 (adapted from [23]).

Regarding *Informed by risk management*, obstruction risk was analyzed, and, similarly to the IRO graphic, basin BD 9TRO022 presents the highest value by a large difference (Figure 31).

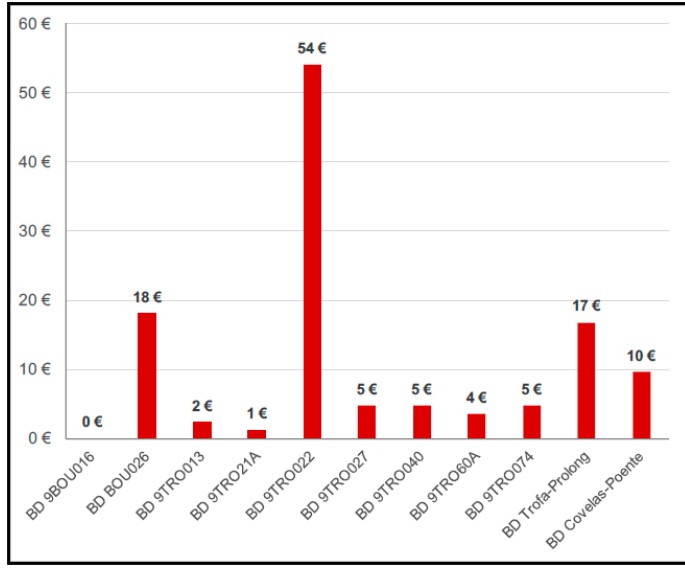

**Figure 31.** Obstruction risk in SAR-Trofa in 2017–2018 (adapted from [23]).

The methodology of GOIH proposes several actions according to each stage of the exploration section.

Operational control—inflows control; prevention planning; structuring functional teams; incorporation of all operational actions in one single tool.

Occurrence response—reduce response time (by optimizing the identification and verification process and with systematic monitoring and control of external providers); define an operational strategy, prioritizing service continuity and damage minimization.

Information: Real Data—robust databases and systematic analysis of the operational records.

Informed by risk management—treatment of exploration real data (risk of inflows, collapses, flooding, and service interruption); organization of actions and pre-positioning of operational teams according to the different risks of the drainage basins.

The particular focus of the exploration strategy is inflows. The following figure (Figure 32) shows the volume evolution for delivered water at high levels, accounted water and not accounted water, for the Trofa municipality, for SAR-Trofa, and for the 2 drainage basins that have flow meters available.

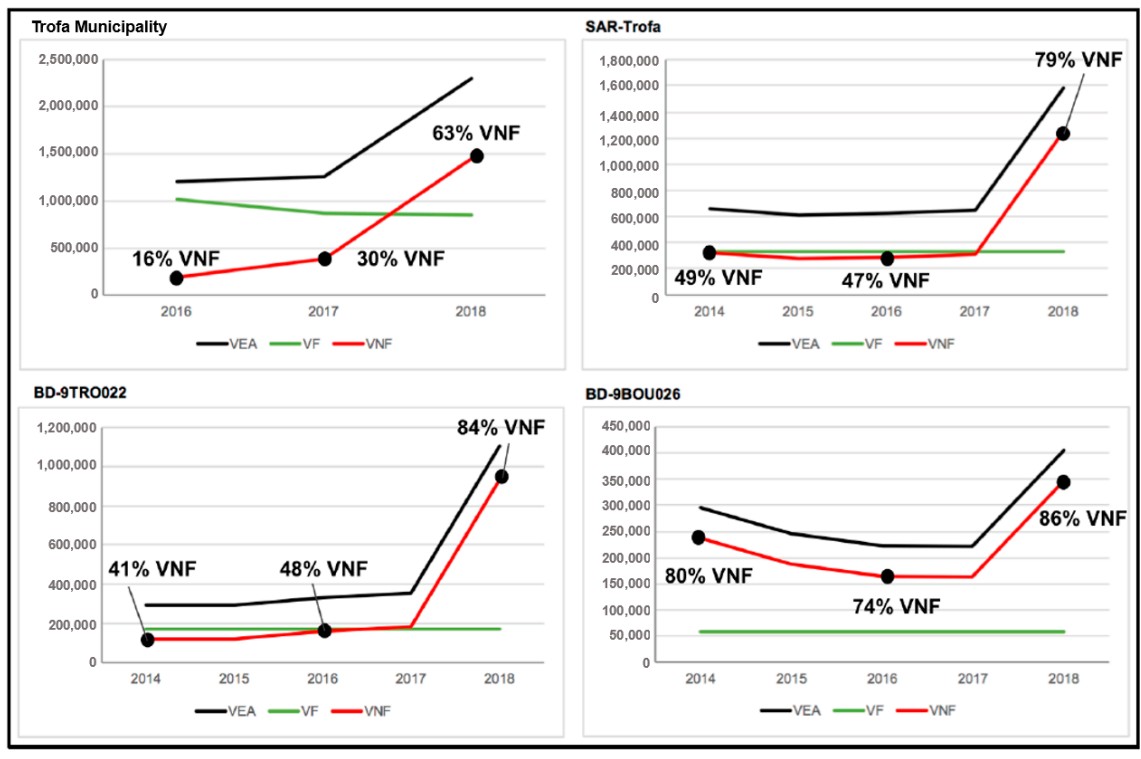

**Figure 32.** Annual variation of volumes delivered at high levels: Accounted Volumes and Not Accounted Volumes (adapted from [23]).

The difference between delivered wastewater and accounted volumes is notable, and in some cases, the first is 80% higher than the last. It appears that, even in a scenario that ensures an increase in accounted volumes, the best client's record, and more rigorous billing, the difference between delivered wastewater and accounted volumes would still be significantly high.

The delivered wastewater volumes can be influenced by rainfall volumes, and the following graphics show this. It is noticeable that the drainage basin BD 9BOU026 is specially influenced by these inflows because the delivered wastewater curve accompanies the precipitation curve very similarly (Figures 33 and 34).

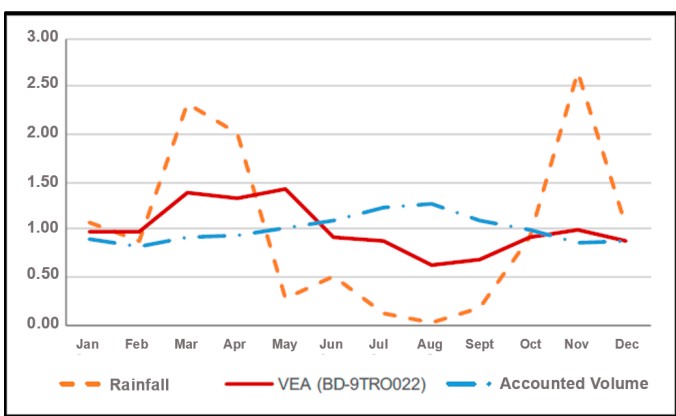

**Figure 33.** Monthly variation of delivered volumes (VEA), billed volumes (VF), and total precipitation in drainage basin BD 9TRO022 (adapted from [23]).

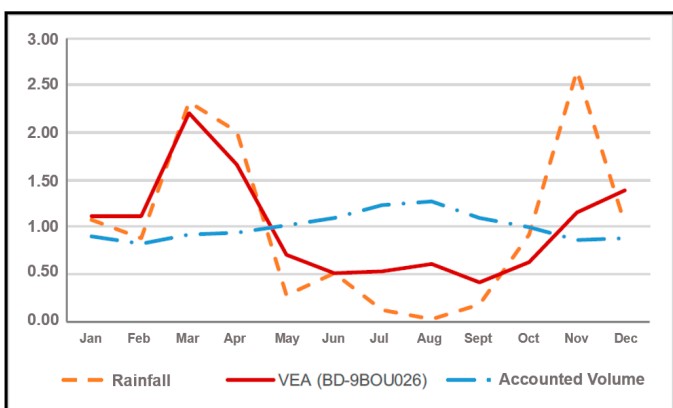

**Figure 34.** Monthly variation of delivered volumes (VEA), billed volumes (VF), and total precipitation in THE drainage basin BD 9BOU026 (adapted from [23]).

### 4.2.3. Intervention

Regarding *Monitoring* intervention actions, the following are suggested: lifting installations, critical sites of the system, volumes at existing measurement points, and gas measurement campaigns.

*Maintenance* is proposed for lifting installations and manholes.

As for *Rehabilitation* and *Reinforcement*, the methodology suggests small interventions of rehabilitation in collectors, contract work in different streets in Santiago do Bougado, and manhole rehabilitation. Organizing the data given by the management entity during these actions is part of the *Reinforcement* action.

### 4.2.4. Decision Support

This section is divided into three strategies:

Investment strategy—global investment to make in SAR-Trofa, by drainage basin, based on the assessment of hydraulic infrastructures and IRO.

Operational strategy—establishing investment scenarios, accounting for the operational return.

Specific decision strategy—analysis of specific situations related to the control of inflows.

Regarding the investment strategy, the real data available only allowed for the calculation of the risk of obstruction, so the index directly associated with risk (Risk Management Index) was not considered because it would have influenced the investment on one risk only (Table 11).

**Table 11.** Summary of assessment indexes for SAR-Trofa, relating to 2019 (adapted from [23]).

| Drainage Basin | Status Assessment Index | Performance Assessment Index | Risk Assessment Index | Infrastructural Assessment Index |
|---|---|---|---|---|
| BD 9BOU016 | 0.08 | 0.07 | 0.06 | 0.07 |
| BD BOU026 | 0.12 | 0.13 | 0.12 | 0.12 |
| BD 9TRO013 | 0.08 | 0.07 | 0.08 | 0.07 |
| BD 9TRO21A | 0.11 | 0.12 | 0.08 | 0.11 |
| BD 9TRO022 | 0.10 | 0.10 | 0.12 | 0.11 |
| BD 9TRO027 | 0.08 | 0.09 | 0.09 | 0.09 |
| BD 9TRO040 | 0.09 | 0.07 | 0.10 | 0.09 |
| BD 9TRO60A | 0.08 | 0.07 | 0.07 | 0.07 |
| BD 9TRO074 | 0.09 | 0.08 | 0.10 | 0.09 |
| BD Trofa Prolongamento | 0.08 | 0.09 | 0.09 | 0.09 |
| BD Covelas Poente | 0.10 | 0.09 | 0.09 | 0.09 |

The operational strategy establishes hypothetical scenarios for the reduction of non-revenue volumes, analyzing:

Estimate the investment scheduled by the managing entity—EUR 54,749 for infrastructure rehabilitation and EUR 67,594 for reinforcement.

Estimate of the proposed investment through GOIH—EUR 100,000 for rainfall volume control and EUR 75,000 for reinforcement of human resources.

Annual gains simulation for different scenarios (Table 12).

**Table 12.** Summary of six hypothetical scenarios of VEA decrease in SAR-Trofa (adapted from [23]).

| | Investment | | | | | Annual Gain | Return | |
|---|---|---|---|---|---|---|---|---|
| | Rehabilitation | Reinforcement | Volumes Control | Operational | Total | (Next Year) | (Years) | (Months) |
| ↓ 40% VEA | | | | | | EUR 374,932 | 0.89 | 11 |
| ↓ 50% VEA | | | | | | EUR 468,665 | 0.71 | 9 |
| ↓ 60% VEA | EUR 54,749 | EUR 67,594 | EUR 100,000 | EUR 112,500 | EUR 334,843 | EUR 562,398 | 0.60 | 7 |
| ↓ 70% VEA | | | | | | EUR 656,131 | 0.51 | 6 |
| ↓ 75% VEA | | | | | | EUR 702,998 | 0.48 | 6 |
| ↓ 80% VEA | | | | | | EUR 749,864 | 0.45 | 5 |

Non-revenue volume simulation for different scenarios (Table 13).

**Table 13.** Summary of three scenarios of annual variation of delivered volumes and non-revenue volumes in Trofa (adapted from [23]).

| Drainage Basin | ↓ 60% VEA | | | ↓ 70% VEA | | | ↓ 80% VEA | | |
|---|---|---|---|---|---|---|---|---|---|
| | VEA | VNF | | VEA | VNF | | VEA | VNF | |
| BD 9BOU026 | 159,520 | 117,100 | 73% | 119,640 | 77,220 | 65% | 79,760 | 37,340 | 47% |
| BD 9TRO022 | 442,200 | 270,432 | 61% | 331,650 | 159,882 | 48% | 221,100 | 49,332 | 22% |
| Total SAR-Trofa | 680,469 | 348,429 | 51% | 530,039 | 197,999 | 37% | 379,608 | 47,568 | 13% |
| Total Trofa | 1,397,874 | 547,842 | 39% | 1,247,444 | 397,412 | 32% | 1,097,013 | 246,981 | 23% |

Estimate of the return on the operational strategy.

The control of undue rainfall may involve implementing rainfall detour solutions through the use of rainfall control chambers, discharge channels, float or guillotine flow regulating valves, or vortex valves.

The drainage basin BD 9TRO022 is relevant for the more efficient management of SAR-Trofa. An investment of EUR 50,000 in this basin could decrease VEA by 40% to 80% and would have a return period of 1 to 2 months.

Drainage basin BD 9BOU026 is also important to manage regarding these same inflows, and an investment of EUR 50,000 could decrease VEA by 40% to 80%, with a return period between 3 and 6 months.

## 5. Conclusions

Currently, urbanization is being established in concentrated centers, while rural areas are getting deserted. This influences the progress attained in the last few decades regarding the coverage of wastewater drainage systems, water quality in source and receptor media, and the adequate management of urban waters.

The methodology of GOIH is a strategic and broad approach to the existing patrimony, envisioning the optimization of water systems. Following a common methodology with different scope and transversality. It aims to be a process to boost an integrated transformation with fast results that can finance the project further.

SAA-Arouca showed improvement after the application of GOIH, and it is expected to continue to improve as the methodology continues to be implemented. Tackling water losses in the municipality of Arouca has been a clear focus in recent years. The proposed actions implemented with the use of decision support tools have resulted in a significant reduction of water losses in the Arouca Water Supply System (SAA-Arouca).

As for SAR-Trofa, the application of GOIH showed potential improvement if the suggested investment were to take place, particularly focusing on rainfall inflows, which are a significant concern in this system.

The study focuses on operational costs and gains. The main limitation to the study presented is the difficulty of including an integrated economic analysis that cuts across all areas of the hydraulic infrastructure management organization, which could also be a future development.

In the future, this methodology may continue to be tested and perfected, for example in wastewater treatment plants and submarine outfalls, as well as rainfall water drainage systems. And other methods can be added, such as water system modeling and geographical information systems.

To summarize the methodology presented, the work carried out, and the results obtained, these key observations are highlighted:

- The proposed methodology combines asset management, risk management, and technical management concepts and is organized into three main operational areas: assessment, operation, and intervention.
- The operational management strategy was designed with a focus on reducing operating costs, increasing service quality, and safeguarding the environment.
- It was applied to the case studies with the aim of evaluating the methodology and obtaining results to support operational management, particularly in terms of reducing water losses and controlling undue rainfall.
- The need to train human resources was identified as part of an integrated commitment to preserving existing know-how and continuous improvement.
- The pertinence of channeling operational gains into the enhancement of infrastructure and investment in innovation is highlighted.
- In a context of increasingly complex problems and ever more demanding challenges, the contribution of knowledge production in the area of GOIH could provide methodologies that help to ask the right questions, point out the appropriate paths, and facilitate the decision-making process.

**Author Contributions:** Conceptualization, J.C.-G.; Investigation, J.C.-G.; Supervision, J.T.-V. All authors have read and agreed to the published version of the manuscript.

**Funding:** This research received no external funding.

**Data Availability Statement:** The data presented in this study are openly available in https://hdl.handle.net/10216/124498.

**Conflicts of Interest:** The authors declare no conflict of interest.

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
