# Peer review of "A Framework for Operational Management of Urban Water Systems to Improve Resilience"

_water, doi:10.3390/w16010154_

Round 1

Reviewer 1 Report

Comments and Suggestions for Authors

Dear Authors

Thank you for working on an interesting issue on "A Framework for Operational Management of Urban Water 2 Systems to Improve Resilience". Water is an important issue and its management in the urban areas are becoming a challenge. Some of my comments to improve the manuscript are:

1. Introduction section needs to reorder and rearrange completely. For example, merge all the sub-section and make it one section i.e. Introduction. I don't see the subsection makes any sense here. 

2. Similarly, data analysis in the introduction section won't make any sense. More details on Infrastructure data, Exploration and management data are needed to present in the manuscript in a separate section.  In addition to that, objective, subjective and modelling based approach didn't make any meaning in the introduction section and it need to extend these approaches in detail.

3.   Figure 1 and is fine and make sense, but the author needs to improve "The methodology aims for the development of strategies that consider matters of" afterwards and need to rewrite completely. 

4. In section 3, there are several lacunae in the manuscript. line 276-282 can be merged in a sentence or it won't make sense.

5. There are two Figure 4, 5, and 6 that is confusing and misleading. Please recheck.  Line 327-324 should be merged. There are unnecessarily many Figures which can be reduced.    

6. Section 4, application and methodology is lacking. 

7. Decision support is not clear and need some clarification and literature.

8. Some of the literature to add:

https://doi.org/10.1016/j.gsd.2023.100923 

https://doi.org/10.1016/j.jclepro.2017.07.024

I wish author the good luck.

Author Response

The notes of revision were taken into account and improvements were included.

The article was adjusted considering the several noted aspects:

  • Introduction rearrangement ;
  • Better explanation of objective, subjective and modelling based approach;
  • The text after "The methodology aims for the development of strategies that consider matters of" was merged and improved;
  •  Lacunae found in section three;
  • Merging of lines between 276-282;
  • Rearranging Section 3 allowed the clarification of figures 4, 5, and 6;
  • Some figures were deleted from the manuscript;
  • The decision support is included in the methodology and literature has been reinforced.

Reviewer 2 Report

Comments and Suggestions for Authors

The paper delves into optimizing the management of hydraulic infrastructures that support water supply, wastewater, and stormwater drainage to enhance overall system efficiency. A comprehensive framework for the operational management of urban water systems is proposed, with a specific emphasis on robust management strategies to bolster system resilience. The methodology integrates key concepts from asset management, risk management, and technical management and is organized into three distinct operational areas: Assessment, Operation, and Intervention. The overarching objective is to enhance the efficiency of entities involved in managing water systems. The proposed methodology has been applied in two cases: the Arouca Water Supply System (SAA-Arouca) and the Trofa Wastewater Drainage System (SAR-Trofa), both under the management of Águas do Norte, S.A. Notably, the implementation of the methodology in the SAA-Arouca case resulted in significant reductions in system input volume and pipe busts. For the SAR-Trofa case, the focus was on addressing improper inflows, with proposed solutions aimed at reducing volumes collected by drainage networks and delivered to high-level infrastructures.

While the paper is well-written and engaging, there are a some minor points related to the methodology that warrant consideration:

The authors are encouraged to provide a more detailed explanation of the context leading to the findings in the introduction section, offering readers a clearer understanding of the research's background and significance. In addtion, pls add some refrs.

It would be beneficial to include a discussion of the limitations of the study, providing a balanced perspective on the scope and potential constraints of the proposed methodology.

The conclusion section could be enhanced by rephrasing it to highlight key observations derived from the research. This would provide a more impactful summary of the study's outcomes and their implications.

These suggested modifications aim to strengthen the overall clarity and impact of the paper, ensuring that readers gain a comprehensive understanding of the research and its contributions to the field.

Comments on the Quality of English Language

Moderate editing of English language required

Author Response

All notes were taken into account and the improvements were included, namely:

  • a more detailed explanation in the introduction section;
  • more bibliographic references;
  • discussion of the limitations of the study;
  • rephrasing the conclusion into highlight key observations.

Reviewer 3 Report

Comments and Suggestions for Authors

There is a potential to publish the contents included in this paper. However, the structure and presentation of this paper requires a substantial improvement before the paper can be recommended for publication. 

First, the Introduction chapter needs restructure. Section 1.2 is "introduction of the framework" and the paper title is "a framework ...". Are these two frameworks the same?  Furthermore, Section 1.3 is not appropriate for inclusion in this chapter.

Second, the authors need include an appropriate literature review in the paper.

Third, the authors must clearly present the original materials for this papers. It seems most figures and tables in this paper were taken from previously published materials. The authors need properly explain all figures and tables in the text.

Author Response

Regarding the notes about the Introduction and Section 1.3, the revisions were included as suggested.

Also all figures and tables are now mentioned/explained in the text.

The note regarding literature review was also taken into account.

As for the figures and tables sources, we wish it would be considered that the present study was done in the context of a PhD thesis. The aim of this article is to summarize those results, as well as present updated inputs, as shown in figures 23, 24, and 27.

Round 2

Reviewer 3 Report

Comments and Suggestions for Authors

I am happy for the paper to be accepted in its present version.